# MAR: Medical Asymmetric Retriever for Efficient Chinese Medical Dense Retrieval

## Abstract

Embedding models are critical for domain-specific information retrieval (IR), particularly in healthcare, where accurate, low-latency access to medical knowledge can enhance clinical decision support and mitigate hallucinations in retrieval-augmented generation (RAG) systems. However, Chinese medical retrieval remains underdeveloped due to the absence of high-quality medical retrieval benchmark. To address this limitation, we propose a novel high-quality Chinese **Med**ical **T**ext **E**mbedding **B**enchmark (**MedTEB**), which covers three practical tasks close to real-world scenarios: retrieval, reranking, and semantic textual similarity (STS). We introduce comprehensive LLM-based annotation in the construction process to improve the quality of curated datasets. Through evaluating existing powerful general-purpose embedding models on MedTEB, we demonstrate that MedTEB is a challenging domain-specific embedding benchmark to evaluate models' retrieval capabilities on Chinese medical retrieval. On this foundation, we propose **M**edical **A**symmetric **R**etriever (**MAR**), an asymmetric embedding architecture that decouples query and document encoding: a lightweight encoder handles online queries with minimal latency, while a powerful and offline LLM-based encoder preserves retrieval quality. Optimizing the asymmetric architecture brings to new challenges. We introduce a novel two-stage optimization framework: 1) **query encoder alignment** and 2) **joint fine-tuning**. Through the novel approach, MAR achieves state-of-the-art (SOTA) performance on MedTEB while maintaining lightweight inference speeds comparable to small-size BERT-style embedding models, leading to an excellent trade-off on accuracy and efficiency and thus offering a practicable and effective solution for real-world Chinese medical retrieval scenarios. Our code, data and model will be made publicly available to facilitate future research on domain-specific IR.

## 1 Introduction

Embedding models have become the backbone of modern natural language processing (NLP), facilitating tasks such as retrieval, reranking, and classification (Reimers & Gurevych, 2019). Their role is crucial in retrieval-augmented generation (RAG) systems (Lewis et al., 2020), which leverage external knowledge to enhance large language models (LLMs). In specialized domains such as healthcare, where LLMs often lack deep expert knowledge, accurate and low-latency access to medical knowledge can enhance clinical decision support and mitigate hallucinations in RAG, making domain-specific, low-latency embeddings indispensable.

Despite recent rapid progress in general-domain embedding models (e.g., BGE (Chen et al., 2024a), GTE (Li et al., 2023), Qwen3-Embedding (Zhang et al., 2025)), Chinese medical text embedding has received limited attention. Existing benchmarks like C-MTEB (Xiao et al., 2024) include only two Chinese medical retrieval datasets, but both exhibit annotation noise and false negatives (see Appendix E). Moreover, current powerful embedding models are mostly LLM-based (Lee et al., 2024), delivering strong performance but at the expense of substantial latency and computational overhead, which limit their applications in latency-sensitive senarios such as real-time medical QA. This highlights the challenge of balancing performance and deployability.

To address the critical gap in standardized evaluation for Chinese medical text embedding, we introduce the Chinese **Med**ical **T**ext **E**mbedding **B**enchmark (**MedTEB**), which consists of three newly

curated tasks—retrieval, reranking, and medical synonym STS—along with two existing public datasets. We employ a comprehensive LLM-based annotation pipeline to improve label quality. Evaluations show that even powerful general-purpose embedders underperform on MedTEB, confirming its difficulty and domain-specificity.

Building on this foundation, we propose **M**edical **A**symmetric **R**etriever (**MAR**), an asymmetric embedding architecture that decouples query and document encoding: a lightweight query encoder serves online requests with minimal latency, while a more powerful, offline document encoder preserves retrieval quality (Wang & Lyu, 2023). To progressively bridge the two encoders and directly optimizes retrieval, we introduce a two-stage optimization framework: 1) query encoder alignment and 2) joint fine-tuning. As shown in Figure 1, while most embedding models exhibit a clear accuracy-latency trade-off, MAR breaks this trend. It matches the retrieval accuracy of heavyweight LLM-based embedding models while sustaining QPS levels comparable to small-size BERT-style embedding models. We further observe that as the document encoder scales up, the asymmetric model progressively closes the gap with LLM-based embedding models, offering a practical path to scale retrieval performance without sacrificing latency.

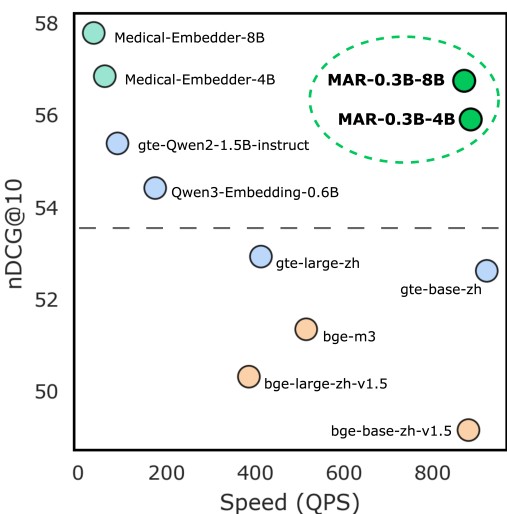

Figure 1: Efficiency-performance trade-off on MedTEB Retrieval. The x-axis shows queries per second (QPS) on a single A100 80GB GPU; the y-axis reports nDCG@10 of MedTEB Retrieval.

The primary contributions of our work are as follows:

• We introduce MedTEB, a comprehensive benchmark for Chinese medical text retrieval, establishing a reliable standard for future domain-specific evaluation.

• We propose MAR, an asymmetric embedding model for the Chinese medical domain that achieves SOTA performance with low inference latency.

• We open-source benchmark, models, and code to foster future research in domain-specific retrieval.

## 2 RELATED WORK

**Embedding Models.** Text embedding models have advanced rapidly alongside pretrained language models. Early works such as Contriever (Izacard et al., 2021) explored unsupervised contrastive pretraining, while more recent models like E5 (Wang et al., 2022), GTE (Li et al., 2023), and the BGE series (Chen et al., 2024a) leveraged large-scale contrastive pretraining to obtain strong general-purpose embeddings. In the biomedical domain, specialized models such as MedCPT (Jin et al., 2023) and BMRetriever (Xu et al., 2024) leverage large-scale medical corpus and tuning language models for enhanced retrieval. Recently, decoder-only embedding models such as Qwen3-Embedding (Zhang et al., 2025), bge-en-icl (Li et al., 2024a), and NV-Embed (Lee et al., 2024) have achieved state-of-the-art performance on MTEB (Muennighoff et al., 2022).

Despite these advances, most LLM-based models contain billions of parameters. While they deliver strong accuracy, their high latency and computational overhead make them impractical for latency-sensitive applications such as real-time medical retrieval. This gap highlights the urgent need for lightweight yet effective embedding models in specialized domains.

**Medical Embedding Benchmarks.** MTEB (Muennighoff et al., 2022) provides a comprehensive benchmark across languages and tasks, and its Chinese extension C-MTEB (Xiao et al., 2024) in-

cludes several Chinese embedding model datasets. Recent work such as R2MED (Li et al., 2025) has introduced benchmarks for reasoning-driven medical retrieval. However, domain-specific evaluation in the Chinese medical domain remains scarce. Existing medical related benchmarks, CmedqaRetrieval (Zhang et al., 2017), MedicalRetrieval (Long et al., 2022), and reranking datasets such as CMedQA-v1 and CMedQA-v2 (Zhang et al., 2018) are all included in C-MTEB. However, the retrieval tasks suffer from annotation noise and false negatives, leaving only the reranking tasks relatively reliable. As a result, the field still lacks comprehensive, high-quality benchmarks for Chinese medical text embedding, leaving a major gap for developing and evaluating domain-specific embedding models.

**Asymmetric architecture** A growing number of work explores asymmetric embedding architectures to improve retrieval efficiency. These can be broadly categorized into two families. (1) Pruning and distillation approaches: Works such as KALE (Wang & Lyu, 2023; Campos et al., 2023) prune layers from a BERT-based large encoder to initialize a lightweight query encoder, then apply alignment losses such as Euclidean distance or KL divergence to distill knowledge from the teacher. (2) Heterogeneous encoder approaches: Other works, including ScalingNote (Huang et al., 2024) and HotelMatch (Askari et al., 2025), align query and document encoders with different architectures or modalities. Our approach differs in three ways: (i) we use a decoder-only document encoder that supports heterogeneous alignment, (ii) we introduce a two-stage alignment framework to progressively bridge query and document encoders and directly optimize retrieval, and (iii) unlike Hotel-Match, which projects query embeddings to a higher dimension through an additional linear layer, leading to higher retrieval cost, our design removes this projection, keeping the original lightweight dimension and yielding a simpler and more efficient architecture.

## 3 MEDTEB

Chinese medical text embedding benchmarks remain scarce. Among the few available benchmarks, CmedqaRetrieval (Zhang et al., 2017) and MedicalRetrieval (Long et al., 2022) are well known and widely used. These datasets are constructed primarily from human-labeled query-answer pairs sourced from online medical Q&A platforms, such as patient inquiries and physician responses. However, this methodology inherently ignores potentially relevant yet unlabeled candidate answers associated with other pairs. The medical domain further exhibits *topic intensity*: common diseases or medications often generate a large volume of semantically similar queries and answers, amplifying the risk of false negatives (See Appendix E for examples).

To empirically assess this issue, we performed preliminary LLM-assisted annotation on both benchmarks and report detailed findings in Appendix E. Our analysis indicates that, on average, each query in MedicalRetrieval is associated with approximately 8.6 candidate passages labeled as negative but suggested by the LLM as potentially relevant; for CmedqaRetrieval, this figure rises to approximately 19. We emphasize that these results are preliminary and do not imply ground-truth correctness, as the LLM's judgments may contain errors, but the scale of flagged negatives strongly suggests systemic annotation gaps in current benchmarks.

To address these shortcomings, we construct MedTEB, a benchmark featuring three new tasks: Retrieval, Reranking, and Synonym STS. We also incorporated two high-quality, human-verified existing public datasets (CMedQAv1-reranking (Zhang et al., 2017) and CMedQAv2-reranking (Zhang et al., 2018), both).

### 3.1 CONSTRUCTION METHOD

**Retrieval** Prior studies like AIR-Bench (Chen et al., 2024b) and Thomas et al. (2024) demonstrate the reliability of LLM-generated relevance labels in information retrieval benchmark. Building on this, we adopt a multi-LLM labeling pipeline. We curate an anonymized Chinese medical corpus $\mathcal{D}$ from publicly available resources and collect real-world, anonymized user queries $\mathcal{Q}$ from our online service. For each query $q_i \in \mathcal{Q}$, a candidate pool of documents is retrieved by multiple retrieval models and then labeled by multiple LLMs. The final retrieval dataset comprise a query set $\mathcal{Q}$, a labeled corpus $\mathcal{D}' \subseteq \mathcal{D}$, and relevance labels $\mathcal{R} = \{(q_i, d_j, y_{ij}) \mid y_{ij} \in \{0, 1\}\}$. Compared with AIR-Bench, our pipeline differs in three respects: (i) it targets the medical domain, (ii) it uses real-world queries rather than synthetic ones, (iii) it employs multiple LLMs and a large multi-

retriever candidate pool to mitigate both mislabelled negatives and unlabeled false negatives. The detailed pipeline and anonymization steps are provided in Appendix B.

**Rerank** We use the same multi-LLM consensus annotation as in Retrieval. For each query $q_i \in \mathcal{Q}$, we derive positives $P_i = \{d_j \in \mathcal{D}' : y_{ij} = 1\}$ and negatives $N_i = \{d_j \in \mathcal{D}' : y_{ij} = 0\}$. The reranking dataset is a collection of triplets $\mathcal{T}_{\textbf{Rerank}} = \{(q_i, \mathcal{P}_i, \mathcal{N}_i)\}$, where $\mathcal{P}_i$ is a list sampled from $P_i$ and $\mathcal{N}_i$ is a list sampled from $N_i$.

**STS** We first build a medical synonym dictionary with domain experts. For each $q_i \in \mathcal{Q}$, GPT-4o generates three sentences: a positive $s_i^+$ (synonym substitution with semantics preserved), a hard negative $s_{i,1}^-$ (synonym substitution with semantics changed), and an easy negative $s_{i,2}^-$ (no synonym substitution with semantics changed). We then sample $s_i \in \{s_i^+, s_{i,1}^-, s_{i,2}^-\}$ and pair it with $q_i$ to form $(q_i, s_i, y_i)$, where $y_i = \mathbf{1}[\, s_i = s_i^+ \,] \in \{0, 1\}$. The dataset is $\mathcal{T}_{\text{STS}} = \{(q_i, s_i, y_i)\}$, evaluating fine-grained synonym understanding.

## 3.2 EVALUATION OF EXISTING EMBEDDING MODELS

The statistics of MedTEB are summarized in Table 1. Average results of CMedQA (CMedQAv1-reranking and CMedQAv2-reranking) and new tasks (Retrieval, Rerank and STS) of existing general-domain embedding models are shown in Table 2, (full zero-shot average results shown in Table 9, and detailed statistics shown in Appendix G), and we also compute the Spearman rank correlation coefficient (Spearman, 1961) between their rankings of average scores on CMedQA and new tasks. Results shows that there is a great gap of the performance between CMedQA and new tasks by existing general domain embedding models (CMedQA Average scores: 85.15 vs. 57.85 of new tasks), showing the challenging of new medical tasks and the underdevelopment of embedding models in medical domain. The Spearman rank correlation coefficient is 0.354 with p-value 0.215 ($\gg$ 0.05), indicating that our new tasks are not redundant with existing medical tasks, but rather explore the model's performance in the

Table 1: MedTEB statistics

| Task | Test | Train | Main Metric |
|---|---|---|---|
| *New tasks* | | | |
| Retrieval | 734 | | NDCG@10 |
| Rerank | 1,128 | 20,000 | MAP |
| Synonym STS | 5,000 | 10,000 | Pearson |
| *Public datasets* | | | |
| CMedQA-v1-rk. | 1,000 | | MAP |
| CMedQA-v2-rk. | 1,000 | 50,000 | MAP |

Table 2: Average performance on CMedQA vs. MedTEB New Tasks, together with Spearman correlation ($\rho$) and p-value.

| Benchmark | Avg Score | Spearman (p-value) |
|---|---|---|
| CMedQA | 85.15 | 0.354 (0.215) |
| New Tasks | 57.85 | |

medical field from fresh perspectives. Notably, decoder-only models like Qwen3-Embedding achieve the strongest performance (Qwen3-Embedding-8B achieves average scores 64.52 on new tasks), but their high latency and computational cost limit real-world applicability. MedTEB thus provides a more rigorous and realistic benchmark for evaluating medical text embeddings, highlighting both the limitations of current models and the need for efficient, domain-specialized solutions.

## 4 MEDICAL ASYMMETRIC RETRIEVER

Given the limitations of existing models on MedTEB (as shown in Section 3.2), we propose a novel training framework of **Asymmetric embedding architecture** to improve Chinese medical embedding models. As illustrated in Figure 2, the document encoder processes the entire corpus offline to build a vector index, while the query encoder operates online, encoding user queries for efficient retrieval. In this section, we describe our high-quality training data construction for medical domain, and a two-stage training strategy designed for our asymmetric embedding architecture.

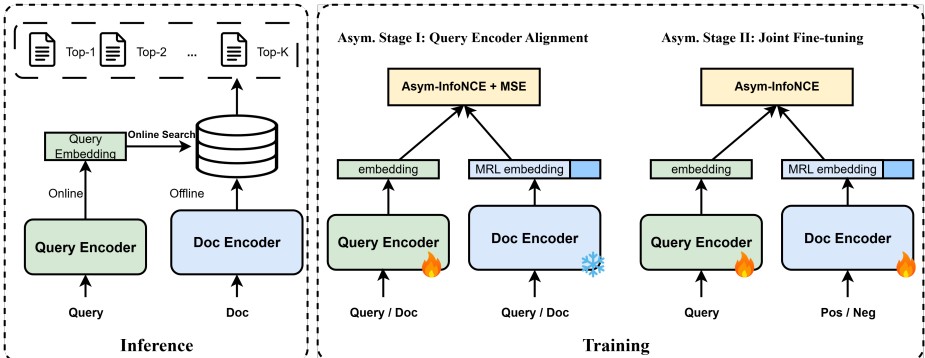

Figure 2: Inference and Training pipeline for asymmetric embedding model. Stage I: Query encoder is trained to align with the frozen document encoder using Asym-InfoNCE and MSE losses. Stage II: Both encoders are jointly fine-tuned with Asym-InfoNCE loss on retrieval data.

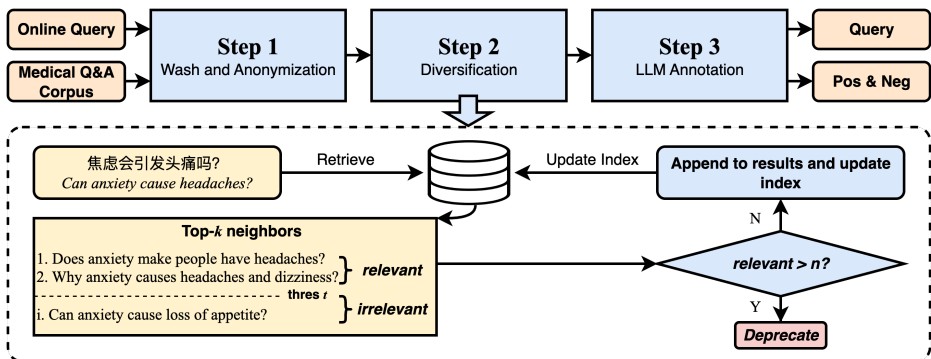

Figure 3: Pipeline for constructing high-quality medical retrieval data. The process includes (1) wash and anonymization of raw queries and documents, (2) deduplication and diversification via embedding-based similarity filtering, and (3) LLM-based labeling of positive and negative samples to mitigate false negatives.

## 4.1 HIGH-QUALITY DATA CONSTRUCTION

The quality of negative samples, especially *hard negatives* plays a critical role in training effective embedding models. However, in the medical domain, the *topic intensity* phenomenon (as discussed in Section 3) results in many queries having a large number of potential positives. This abundance of hidden positives undermines conventional hard negative mining: Top-$k$ retrieval often introduces false negatives due to many unlabeled but relevant documents, threshold-based filtering suffers from blurred decision boundaries, and LLM-based annotation becomes prohibitively expensive when applied to such large candidate pool. To address this issue, we design a **diversity-aware data curation pipeline** that reduces redundancy and improves annotation reliability through three key steps (Figure 3): (i) collect and anonymize a broad Chinese medical corpus from publicly available resources and real queries from our online service; (ii) deduplicate/diversify queries and corpus via a dynamic vector-index filter to reduce semantic redundancy; and (iii) use GPT-4o to label reliable positives/negatives from top-50 retrieved candidates, producing 500K triples $(q, d^+, d^-)$. Implementation details are in Appendix C.

Additionally, to improve query–document alignment in asymmetric models, we construct a **query alignment dataset**. Each query is paired with itself as the positive document, while in-batch samples serve as negatives. In total, we generate 2.8M query-side triples $(q, q, q^-)$ and 5.6M document-side triples $(d, d, d^-)$ for this task, which will be used in Section 4.3.1.

## 4.2 INDEPENDENT INITIALIZATION

We firstly independently train a symmetric dual-tower model to inject domain-specific knowledge into both the query and document encoders. This initialization step establishes a strong foundation for the subsequent asymmetric alignment phase.

**Query Encoder** Following prior work (Chen et al., 2024a; Xiao et al., 2022), we adopt a three-stage training pipeline for the query encoder: (1) RetroMAE pretraining (Xiao et al., 2022); (2) Unsupervised pretraining with InfoNCE loss (Oord et al., 2018); (3) Supervised fine-tuning with InfoNCE loss. Details of query encoder training are described in Appendix D.1.

**Document Encoder** For the document encoder, we fine-tune a large pretrained language model with LoRA (Hu et al., 2022) to reduce compute while preserving performance (Li et al., 2024a; Wang et al., 2023). For flexible deployment and compatibility with a smaller query encoder's dimension, we adopt Matryoshka Representation Learning (MRL) (Kusupati et al., 2022), training the model to output embeddings at multiple dimensions so we can later select the dimension that matches the query encoder during asymmetric alignment. Details are in Appendix D.2.

## 4.3 ASYMMETRIC EMBEDDING ARCHITECTURE

While LLM-based embedders achieve state-of-the-art retrieval accuracy, their high computational cost and latency are prohibitive for real-time applications. To resolve this trade-off, we propose an **Asymmetric embedding architecture**, which pairs a lightweight query encoder for fast online inference with a powerful document encoder whose embeddings are pre-computed offline. A key challenge, however, is the inherent misalignment between the embedding spaces of these disparate models. We address this by designing a two-stage training strategy: (1) query encoder alignment stage to map the query encoder's space to the document encoder's, followed by (2) joint fine-tuning stage to optimize both for the end retrieval task.

### 4.3.1 ASYMMETRIC STAGE I: QUERY ENCODER ALIGNMENT

To close the semantic gap, we freeze the document encoder (the *teacher*) and update only the query encoder (the *student*). Training uses the **query alignment dataset** (Section 4) with each query or document paired with itself as the positive. We employ a hybrid objective:

**Asymmetric Contrastive Loss** We use Asym-InfoNCE with frozen document encoder as teacher:

$$\mathcal{L}_{\text{Asym-InfoNCE}} = -\log \frac{\exp(s^+/\tau)}{\exp(s^+/\tau) + \sum_{i=1}^{N} \exp(s_i^-/\tau)}, \tag{1}$$

where $s^+ = \text{sim}(E_Q(q), E_D(d^+))$ and $s_i^- = \text{sim}(E_Q(q), E_D(d_i^-))$; $\text{sim}(\cdot, \cdot)$ denotes cosine similarity. $E_Q$ and $E_D$ are the query and document encoders, $d^+$ and $d^-$ are positive and negative documents, and $\tau$ is the temperature. As $q$ and $d^+$ are identical texts in this stage, the loss aligns the student to the teacher in a contrastive manner.

**MSE Loss** For further alignment, we add:

$$\mathcal{L}_{\text{MSE}} = \|E_Q(\textbf{text}) - E_D(\textbf{text})\|_2^2, \tag{2}$$

which penalize the L2 distance between normalized query and document embeddings of the same text to match the teacher's embedding space.

**Final Objective** The overall objective for Stage 2.1 is a weighted combination:

$$\mathcal{L}_{\text{Stage 2.1}} = \lambda_1 \mathcal{L}_{\text{Asym-InfoNCE}} + \lambda_2 \mathcal{L}_{\text{MSE}}, \tag{3}$$

with $\lambda_1 = \lambda_2 = 1$. Asym-InfoNCE provides soft alignment through relative ranking signals, while MSE enforces absolute alignment in embedding space. Together, they guide the query encoder to faithfully approximate the semantic space of the stronger document encoder.

## 4.4 ASYMMETRIC STAGE II: JOINT FINE-TUNING

After alignment, we unfreeze both encoders and perform end-to-end joint fine-tuning. The goal of this stage is to further enhance retrieval performance by jointly optimizing the two encoders to better discriminate between positive and negative documents. We adopt the **Asym-InfoNCE loss** as the sole objective, leveraging hard negatives and in-batch negatives (Xiong et al., 2020; Karpukhin et al., 2020) to enrich the negative samples. This end-to-end optimization directly optimizes the model for the retrieval task. The final models, Medical Asymmetric Retriever (MAR), yielding a strong accuracy–latency trade-off for real-world, latency-sensitive medical retrieval.

## 5 EXPERIMENTS

### 5.1 SETUP

**Models.** We denote our fine-tuned query encoder as **Medical-Embedder-base**, initialized from gte-multilingual-mlm-base (Zhang et al., 2024b). The document encoders are fine-tuned from Qwen3-4B (Yang et al., 2025) and Qwen3-8B, referred to as **Medical-Embedder-4B** and **Medical-Embedder-8B**, respectively. Based on these, we evaluate two asymmetric variants: **MAR-0.3B-4B** (a ∼0.3B query encoder paired with Medical-Embedder-4B) and **MAR-0.3B-8B** (same query encoder with Medical-Embedder-8B). For baselines, since our method targets high-efficiency online deployment, we focus on relatively *lightweight* yet strong baselines that achieve state-of-the-art results on MTEB and are widely used in practice. To this end, we compare against strong open-source embedding models covering both *lightweight* BERT-style encoders and LLM-based embedders with moderate parameter sizes, including the BGE series (Xiao et al., 2024), GTE series (Li et al., 2023), Qwen3-embedding series (Zhang et al., 2025), Conan-embedding-v1 (Li et al., 2024b), and stella-base-zh-v3-1792d (Zhang et al., 2024a).

**Training Data.** All baselines and our models are fine-tuned on the same data for fair comparison. The training corpus includes high quality fine-tuning datasets described in Section 4 as well as the training splits of MedTEB (retrieval, reranking, CMedQA, and Synonym STS). Although some baselines have previously seen CMedQA during pre-training, we explicitly include it to prevent potential performance degradation on this task.

**Implementation Details.** For retrieval evaluation, we use FAISS (Johnson et al., 2019) for efficient nearest-neighbor search over the document corpus. For fair comparison, all symmetric baseline models are fine-tuned for 2 epochs, matching the total exposure of our asymmetric models, which observe the fine-tuning data once during independent initialization and once again during joint fine-tuning. All experiments are conducted on 32×A100-40GB GPUs.

### 5.2 MAIN RESULTS ON MEDTEB

Table 3 presents the evaluation of MAR series on the MedTEB benchmark, alongside strong open-source baselines. We observe two key findings: (1) MAR establishes a new state of the art: the 0.3B–4B variant achieves an average score of 78.13, and the 0.3B–8B variant reaches 78.94 — surpassing the strongest baseline, gte-Qwen2-1.5B-instruct (77.61, a decoder-only model), despite using a much smaller query encoder. (2) Both baseline models and our asymmetric variants exhibit consistent performance scaling with model size: enlarging the document encoder from 4B to 8B improves the average score by 0.81. Critically, these gains incur no additional query-time cost, as the 0.3B query encoder remains unchanged, offering an optimal accuracy–latency trade-off for real-time medical retrieval.

### 5.3 ASYMMETRIC VS. SYMMETRIC ARCHITECTURES

Table 4 compares symmetric and asymmetric architectures. As expected, symmetric large-scale models (4B and 8B) deliver the strongest performance. Our asymmetric design, which combines a lightweight query encoder with a large document encoder, achieves performance that closely approaches the document encoder's upper bound while remaining far superior to the lightweight baseline (symmetric 8B 65.63 vs. asymmetric 8B 65.21). As the document encoder scales from 4B to

Table 3: Results of our models compare to the baselines on MedTEB. Best results in **bold**, second-best in underline. Asymmetric models are marked with †.

| Model | Params (Q/D) | CMed v1 | CMed v2 | Retr. | Rer. | STS | Avg |
|---|---|---|---|---|---|---|---|
| *Baselines* | | | | | | | |
| bge-small-zh-v1.5 | 24M / 24M | 80.21 | 81.69 | 44.33 | 62.30 | 70.50 | 67.81 |
| bge-base-zh-v1.5 | 102M / 102M | 83.37 | 83.31 | 49.16 | 66.73 | 76.24 | 71.76 |
| bge-large-zh-v1.5 | 326M / 326M | 83.23 | 85.15 | 50.32 | 67.55 | 78.95 | 73.04 |
| bge-m3 | 568M / 568M | 82.98 | 83.32 | 51.35 | 66.90 | 78.34 | 72.58 |
| Conan-embedding-v1 | 326M / 326M | **89.89** | 88.77 | 52.75 | 69.31 | 81.49 | 76.44 |
| stella-base-zh-v3-1792d | 102M / 102M | 87.16 | 88.28 | 53.31 | 69.56 | 80.52 | 75.77 |
| gte-multilingual-base | 305M / 305M | 86.21 | 86.37 | 53.37 | 69.38 | 82.36 | 75.54 |
| gte-base-zh | 102M / 102M | 85.31 | 86.44 | 52.62 | 69.35 | 79.73 | 74.69 |
| gte-large-zh | 326M / 326M | 85.44 | 86.97 | 52.93 | 69.97 | 81.48 | 75.36 |
| gte-Qwen2-1.5B-instruct | 1.78B / 1.78B | 87.68 | 87.15 | 55.39 | 72.35 | 85.50 | 77.61 |
| Qwen3-Embedding-0.6B | 596M / 596M | 85.58 | 86.09 | 54.42 | 70.94 | 80.42 | 75.49 |
| *Ours* | | | | | | | |
| MAR-0.3B-4B† | 305M / 4.02B | 86.04 | 87.31 | 55.91 | 72.84 | **88.53** | 78.13 |
| MAR-0.3B-8B† | 305M / 8.19B | 88.34 | **88.86** | **56.75** | **73.67** | 87.07 | **78.94** |

Table 4: Comparison of asymmetric and symmetric embedding architectures.

| Query Encoder | Doc Encoder | Params (Q/D) | Retrieval | Rerank | Avg |
|---|---|---|---|---|---|
| Medical-Embedder-base | | 305M / 305M | 54.16 | 69.63 | 61.90 |
| Medical-Embedder-base | Medical-Embedder-4B | 305M / 4.02B | 55.91 | 72.84 | 64.38 |
| | Medical-Embedder-8B | 305M / 8.19B | 56.75 | **73.67** | 65.21 |
| Medical-Embedder-4B | | 4.02B | 56.85 | 73.26 | 65.06 |
| Medical-Embedder-8B | | 8.19B | **57.79** | 73.47 | **65.63** |

8B, the performance of the asymmetric model increases accordingly (4B score: 64.38 vs. 8B score: 65.21 ), demonstrating the effectiveness of leveraging larger document encoders for improving retrieval accuracy.

## 5.4 ABLATION STUDY

We conduct ablation study on MAR-0.3B-4B except specially mentioned. To better reflect downstream applications, we report performance on both Retrieval and Reranking tasks.

### 5.4.1 TRAINING DESIGN

We conduct experiments on the contribution of different components in our asymmetric training framework. Results are in in Table 5.

For Independent Initialization, removing either query or document encoder initialization leads to severe performance degradation (*w/o query init* scores: 59.66 and *w/o doc init* scores: 50.26 vs. 64.38 for full model). This shows that independent training of both encoders is essential, and stronger symmetric backbones provide a better starting point for asymmetric training.

For Asymmetric Stage, skipping the query alignment stage (*w/o query align* scores: 51.07)

Table 5: Ablation study on training stages and loss functions.

| Setting | Retr. | Rerank | Avg |
|---|---|---|---|
| *Independent Initialization* | | | |
| w/o query init | 50.46 | 68.85 | 59.66 |
| w/o doc init | 37.30 | 63.21 | 50.26 |
| *Asymmetric Stage* | | | |
| w/o query align | 35.34 | 66.79 | 51.07 |
| w/o joint fine-tuning | 42.69 | 68.28 | 55.49 |
| *Loss Design (Asymmetric Stage I)* | | | |
| w/o MSE | 55.19 | 71.94 | 63.57 |
| w/o Contrastive | 55.48 | 72.58 | 64.03 |
| **Full Model** | **55.91** | **72.84** | **64.38** |

Table 6: Impact of different data types in Stage-I on final retrieval performance.

| Training Setup | Stage-I Data | Retrieval | Rerank | Avg |
|---|---|---|---|---|
| Only Stage-I | Fine-tuning data | 44.83 | 69.69 | 57.26 |
| | Query alignment data | 42.69 | 68.28 | 55.49 |
| Stage-I + Stage-II | Fine-tuning data | 49.95 | 71.02 | 60.49 |
| | Query alignment data | **55.91** | **72.84** | **64.38** |

Table 7: Ablation study on alternative approaches to efficient retrieval. For KALE and Wang & Lyu (2023) , we follow their setting by extracting the first three layers of document encoder to initialize query encoder (≈302.8M parameters without LM head, embedding dimension 2560). For ScalingNote, we adopt the same query and document encoder as ours. For the distillation baseline, we use Medical-Embedder-4B as the teacher to provide similarity scores, training the student base with KL-divergence and InfoNCE losses (Ren et al., 2021).

| Model | Asym | Params (Q/D) | Retrieval | Rerank | Avg |
|---|---|---|---|---|---|
| KALE (Campos et al., 2023) | ✓ | 302.8M / 4.02B | 42.67 | 67.42 | 55.05 |
| Wang & Lyu (2023) | ✓ | 302.8M / 4.02B | 39.99 | 66.26 | 53.13 |
| ScalingNote (Huang et al., 2024) | ✓ | 305M / 4.02B | 34.81 | 64.17 | 49.49 |
| Distill-from-4B (Ren et al., 2021) | × | 305M / 305M | 54.68 | 70.76 | 62.72 |
| **MAR-0.3B-4B** | ✓ | 305M / 4.02B | **55.91** | **72.84** | **64.38** |

or the joint fine-tuning stage (*w/o joint fine-tuning* scores: 55.49) also results in clear performance drops (full model 64.38). This confirms our intuition that alignment ensures the student query encoder learns the teacher's embedding space, while joint optimization adapts both encoders to downstream retrieval. Note that our *w/o query align* configuration is close to HotelMatch (Askari et al., 2025), though not identical: HotelMatch applies a linear projection to up-project the small-LM query embeddings to the document encoder dimension and uses separate learning rates for the two encoders, whereas we remove the projection layer and use a single learning rate since we use LoRA to fine-tuning our document encoder.

Finally, we study the loss design in the query align stage. Removing either MSE or Asym-InfoNCE contrastive loss weakens performance. The full model, combining both, consistently achieves the best results. This indicates that both objective contributes to the alignment of embedding space.

### 5.4.2 QUERY ALIGNMENT DATA

We evaluate the role of query alignment data in Stage-I, comparing it with using the Stage-II fine-tuning data for this stage. As shown in Table 6, when only training Stage-I, training with fine-tuning data yields a higher score than with query alignment data (fine-tuning data 57.26 vs. query alignment data 55.49). However, when followed by end-to-end fine-tuning in Stage-II, models initialized with query alignment data achieve a substantially higher final performance (64.38), outperforming those trained with fine-tuning data in Stage-I (60.49). This indicates that query alignment data better prepares the embedding space for downstream retrieval, effectively raising the performance ceiling. In contrast, relying solely on fine-tuning data in Stage-I may lead to premature convergence and suboptimal representation learning.

### 5.4.3 ALTERNATIVE APPROACHES TO EFFICIENT RETRIEVAL

We further compare our two-stage asymmetric training framework against several alternative approaches to efficient retrieval (Table 7). Results show that our proposed method consistently outperforms all alternatives. In particular, asymmetric approaches such as KALE and Wang & Lyu (2023) achieve limited performance (KALE scores: 55.05 and Wang & Lyu (2023) scores: 53.13), for which we assume that their encoder-only training framework has not been fully adapted to decoder-only architectures. Moreover, our method surpasses the distillation baseline (distillation 62.72 vs. ours 64.38), indicating that directly leveraging a large teacher as the document encoder avoids information loss inherent in score distillation and leads to stronger retrieval performance.

## 6 CONCLUSION

In this work, we introduce MedTEB, a new benchmark for Chinese medical text embedder, and propose MAR, an asymmetric model designed for efficient, low-latency medical retrieval. Our architecture, which pairs a lightweight query encoder with a powerful document encoder via a two-stage training strategy, achieves state-of-the-art performance on MedTEB. By releasing the benchmark, models, and training pipeline, we provide both a practical solution for real-world medical RAG systems and a foundation for future research in domain-specific embedding learning. Our future work will include exploring more effective strategies for asymmetric alignment.

## ETHICS STATEMENT

This research has been approved by the National Technology Ethics (Review) Committee. We strictly adhered to ethical guidelines regarding data collection and privacy. User queries were sourced from participants who explicitly consented to a user experience improvement program for non-commercial research. Medical documents were crawled from publicly accessible, non-paywalled websites (e.g., XunYiWenYao[1]) in compliance with robots.txt protocols. We emphasize that these resources are for research purposes only and require rigorous validation before clinical deployment. Expert re-annotation was performed by clinicians from a partner tertiary hospital, compensated at institution-approved rates via official project budgets. MedTEB is released under a CC BY-NC-SA 4.0 license, with model cards explicitly disclaiming diagnostic utility to ensure strict non-commercial, research-only usage.

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

## A  INSTRUCTION

Table 8: Instruction used on MedTEB benchmarks

| Task Name | Instruction Template |
|---|---|
| CMedQAv1-reranking | Based on a Chinese medical question, evaluate and rank the medical information that provide answers to the question. |
| CMedQAv2-reranking | Based on a Chinese medical question, evaluate and rank the medical information that provide answers to the question. |
| MedTEB-Retrieval | Given a Chinese medical question, retrieve medical documents that answer the question. |
| MedTEB-Rerank | Based on a Chinese medical question, evaluate and rank the medical information that provide answers to the question. |
| MedTEB-STS | Retrieve semantically similar text. |

## B  DETAILS FOR MEDTEB

**Retrieval Task Construction.** Given a query $q_i \in \mathcal{Q}$, we used gte-multilingual-base, bge-m3, Conan-embedding-v1 to retrieve and gather a candidate pool of top-500 documents $\mathcal{D}_i = \{d_1, \ldots, d_{500}\}$. Three strong LLMs, DeepSeek-V3 (Liu et al., 2024), Doubao-1.5-Pro (Guo et al., 2025) and GPT-4o (Hurst et al., 2024) then rated each $(q_i, d_j)$ pair on a 5-point relevance scale. To ensure label quality, a document was retained as positive only when all three LLMs agreed, while pairs with partial agreement (only 1 or 2 agreements) were discarded. The final retrieval dataset comprise a query set $\mathcal{Q}$, a refined corpus $\mathcal{D}' \subseteq \mathcal{D}$, and relevance labels $\mathcal{R} = \{(q_i, d_j, y_{ij}) \mid y_{ij} \in \{0, 1\}\}$.

**Detailed Anonymization steps of MedTEB.** All user queries and web documents were processed as follows. 1) Automated PII (personally identifiable information) Detection: We deployed an offline, locally hosted large language model to detect and mask potential PII, including names, locations, phone numbers, and ID numbers. 2) Rule-based Validation: After initial masking, we applied a rule-based validation module to scan residual digits, and keywords. 3) Human Checks: 1% of anonymized data were checked by human, and no re-identifiable content found.

**Detailed zero-shot results on MedTEB.** Table 9 presents the full zero-shot performance of all evaluated models across individual MedTEB tasks. Results show significant performance gaps between general-domain embedders and the medical-specific retrieval challenge. Note that most of baselines have already trained on CMedQA train dataset before.

Table 9: Zero-shot results on MedTEB (%). Best results in **bold**.

| Model | Param. | CMedv1 | CMedv2 | Avg CMed | Retr. | Rerank | STS | Avg. New |
|---|---|---|---|---|---|---|---|---|
| gte-multilingual-base | 305M | 86.11 | 87.40 | 86.76 | 47.80 | 61.51 | 72.39 | 60.57 |
| gte-base-zh | 102M | 86.79 | 87.20 | 86.99 | 44.18 | 58.40 | 75.07 | 59.22 |
| gte-large-zh | 326M | 86.09 | 86.46 | 86.28 | 29.75 | 53.70 | 68.02 | 50.49 |
| gte-Qwen2-1.5B-instruct | 1.78B | 88.16 | 88.12 | 88.14 | 45.14 | 58.99 | 76.81 | 60.31 |
| gte-Qwen2-7B-instruct | 7.61B | 88.20 | 89.31 | 88.76 | 40.94 | 61.07 | 72.67 | 58.23 |
| bge-small-zh-v1.5 | 24M | 77.40 | 79.86 | 78.63 | 35.22 | 55.39 | 57.87 | 49.49 |
| bge-base-zh-v1.5 | 102M | 80.47 | 84.88 | 82.68 | 33.11 | 53.56 | 67.45 | 51.37 |
| bge-large-zh-v1.5 | 326M | 83.45 | 85.44 | 84.45 | 43.05 | 58.31 | 71.90 | 57.75 |
| bge-m3 | 568M | 77.71 | 79.19 | 78.45 | 41.14 | 57.68 | 63.67 | 54.16 |
| Conan-embedding-v1 | 326M | **91.39** | **89.72** | **90.56** | 41.60 | 61.89 | 72.86 | 58.78 |
| stella-base-zh-v3-1792d | 102M | 88.35 | 89.06 | 88.71 | 45.77 | 60.43 | 74.96 | 60.39 |
| Qwen3-Embedding-0.6B | 596M | 80.06 | 81.35 | 80.71 | 47.54 | 64.51 | 68.31 | 60.12 |
| Qwen3-Embedding-4B | 4.02B | 84.43 | 85.06 | 84.75 | 50.14 | **66.67** | **76.49** | 64.43 |
| Qwen3-Embedding-8B | 7.57B | 86.13 | 86.39 | 86.26 | **51.15** | 66.31 | 76.09 | **64.52** |
| Average performance | | 84.62 | 85.67 | 85.15 | 42.61 | 59.89 | 71.04 | 57.85 |
| Spearman Rank Correlation Coefficient (P-value) | | | | | | | | 0.354 (0.215) |

## C    DETAILS FOR HIGH QUALITY DATA CONSTRUCTION

### C.1    DATA CONSTRUCTION

**Data Diversification.**    We apply diversification for query and corpus independently. We first initialized a vector index seeded with 5,000 documents encoded by `gte-multilingual-base`. For each new candidate $x$ (query or document), we retrieve top-$k$ neighbors and discard $x$ if more than $n$ neighbors exceed similarity threshold $t$; otherwise we insert $x$. This is applied separately to queries and corpus, preserving diversity while removing near-duplicates. We summarize the key parameters used during Data Diversification in Table 10, where $k$ represents for top-$k$ retrieved relevant candidates from vector index, $t$ for similarity score threshold, and $n$ for maximum number of related documents.

Table 10: Key parameters used during data generation.

| Parameter | Query | Document |
|---|---|---|
| $k$ (retrieved candidates) | 5 | 5 |
| $t$ (score threshold) | 0.85 | 0.78 |
| $n$ (maximum number) | 1 | 1 |

**LLM annotation.**    For each diversified query $q$, we retrieve top-50 candidates from the diversified corpus and have GPT-4o assign a 5-point relevance score. From scored pools, we select positives and negatives to form triples, yielding 500K fine-tuning instances of triplets $\mathcal{T} = \{(q_i, \mathcal{P}_i, \mathcal{N}_i)\}$, where $\mathcal{P}_i$ is a list sampled from positives $P_i$ and $\mathcal{N}_i$ is a list sampled from negatives $N_i$.

### C.2    ABLATION STUDIES ON DATA DIVERSIFICATION

To evaluate the effectiveness of our diversity-aware data curation pipeline, we conduct an ablation study on the role of query and document-side diversification on Medical-Embedder-base. All configurations use the same amount of training data. As shown in Table 11, the full setting achieves the best performance, demonstrating that both query and document diversification are essential: the former ensures broad topic coverage, while the latter improves the reliability and difficulty of negative samples. This validates the importance of our diversity-aware curation strategy in building high-quality medical retrieval datasets.

Table 11: Impact of query and document diversification on retrieval performance.

| Diversification Setting | Retrieval | Rerank | Avg |
|---|---|---|---|
| w/o query, w/o doc | 51.17 | 68.74 | 59.96 |
| w/ query, w/o doc | 52.23 | 68.98 | 60.61 |
| w/ query, w/ doc | **54.16** | **69.63** | **61.90** |

## D    TRAINING DETAILS OF INDEPENDENT INITIALIZATION

### D.1    QUERY ENCODER TRAINING

**RetroMAE Pretrain.**    We first adopt RetroMAE (Xiao et al., 2022) pretrain, which mask inputs differently in the encoder and a lightweight decoder; the encoder outputs sentence embeddings and the decoder reconstructs the original text via masked language modeling. This stage leverages a 60M unsupervised Medical Q&A corpus.

**Unsupervised Pretrain.**    We perform contrastive unsupervised pretrain using InfoNCE loss (Oord et al., 2018):

$$\mathcal{L}_{\text{InfoNCE}} = -\log \frac{\exp(\mathbf{q}^\top \mathbf{d}^+/\tau)}{\sum_{\mathbf{d} \in \mathcal{D}} \exp(\mathbf{q}^\top \mathbf{d}/\tau)},$$

where $\mathbf{q}, \mathbf{d}^+$ are embeddings of a matched (query, document) pair, $\mathcal{D}$ contains one positive and $|\mathcal{D}| - 1$ negatives, and $\tau$ is a learnable temperature. We use the same unsupervised medical Q&A corpus for RetroMAE pretraining, treating title–content pairs as positives and other documents within the same batch as in-batch negatives.

**Supervised Finetuning.**    The final stage fine-tunes the encoder on high quality fine-tuning datasets described in Section 4 together with the training splits of **MedTEB** (retrieval, reranking, CMedQA, and Synonym STS) using the InfoNCE loss.

## D.2   DOCUMENT ENCODER TRAINING

We fine-tune Qwen3-4B and Qwen3-8B and apply LoRA with rank=32, $\alpha = 64$. We adopt Matryoshka Representation Learning (MRL) (Kusupati et al., 2022), whose training objective aggregates the contrastive loss across this predefined set of dimensions. Specifically, the final loss is the average of the InfoNCE losses computed at each target dimension:

$$\mathcal{L}_{\text{MRL}} = \frac{1}{|M|} \sum_{m \in M} \mathcal{L}_{\text{InfoNCE}}^{(m)},$$
(4)

where $M$ is the set of nested dimensions and $\mathcal{L}_{\text{InfoNCE}}^{(m)}$ is the standard InfoNCE loss calculated using embeddings truncated to the first $m$ dimensions.

## D.3   IMPACT OF PRETRAINING ON QUERY ENCODER

We evaluate the impact of pretraining on the query encoder. As shown in Table 12, combining Retro-MAE and unsupervised domain pretraining achieves the best performance (54.16), outperforming ablated variants. This confirms that multi-stage pretraining enhances the encoder's performance in medical retrieval.

Table 12: Ablation study on pretraining strategies for Medical-Embedder-Base. Combining Retro-MAE and unsupervised domain pretraining leads to the best retrieval performance.

| Training Strategy | Retrieval |
|---|---|
| Finetune only | 52.88 |
| RetroMAE + Finetune | 53.21 |
| RetroMAE + Unsup + Finetune | **54.16** |

# E   ANALYSIS OF OPEN-SOURCE BENCHMARKS

As a preliminary annotation study, we investigate the false negative issue in CmedqaRetrieval and MedicalRetrieval. For each query, we use gte-multilingual-base to retrieve the top-50 candidate documents and re-annotate them using GPT-4o under a 5-point relevance scale with prompt in Table 25.

Results in Table 13 suggest that a large number of retrieved documents, though unlabeled in the original datasets, are judged as relevant by the LLM. Table 14 and Table 15 shows several examples of false negatives and false positives, together with the topic intensity phenomenon in medical domain that certain diseases or drugs generate a large volume of semanti- cally similar queries and answers. This indicates potential annotation incompleteness in existing benchmarks. It is important to note that LLM-based re-annotations are not guaranteed to be fully accurate, and we do not further validate the annotation reliability of GPT-4o in this study. Hence, our findings should be interpreted only as indicative evidence rather than definitive conclusions about dataset quality. Nonetheless, these findings raise concerns about the validity of current benchmarks for a reliable evaluation of medical retrieval capability.

Table 13: LLM re-annotation on open-source medical retrieval benchmarks. To aid interpretation, we assume the LLM labels are pseudo–ground truth. We measure the average number of positive documents per query in the original dataset vs. LLM-labeled data, and identify false positives and false negative.

| Benchmark | Orig. Pos. | LLM-Labeled Pos. | False Positive | False Negative |
|---|---|---|---|---|
| MedicalRetrieval | 0.81 | 9.11 | 0.26 | **8.56** |
| CmedqaRetrieval | 1.42 | 19.94 | 0.46 | **18.98** |

Table 14: An example of false negatives in CmedqaRetrieval.

**Query**
查出说是贫血孩子老烧还有咳嗽
*The child was diagnosed with anemia and has been running a fever with coughing.*

**False Positive (Labeled as positive, but not mention fever and coughing.)**
如果是检查有贫血，可以结合贫血的类型和严重的程度，根据检查结果进行治疗的考虑即可。
*If anemia is detected, treatment can be determined based on the type and severity of anemia, as indicated by the test results.*

**False Negative (Labeled as negative, but annotated as positive by LLM)**
如果只是简单地烧咳嗽，等相应症状，是无需，担心的，但是贫血的原因必须地须要查清楚，一般情况下评选分为营养不良性贫血，还有其他病理性贫血。所以建议到医院进行系统检查，看到底是？出现的什么方面的贫血。然后对症治疗。
*If it is just a simple cough or other corresponding symptoms, there is no need to worry. However, the cause of anemia must be clarified. Generally, anemia is categorized into nutritional deficiency anemia and other pathological anemias. Therefore, it is recommended to go to the hospital for a comprehensive examination to determine the specific type of anemia and then treat it accordingly.*

## F MEDTEB CASE EXAMPLES

We present representative cases from MedTEB tasks in Table 16, Table 17 and Table 18.

## G STATISTICS OF MEDTEB DATASET

For detailed statistics of the MedTEB datasets, please refer to Tables 19 to 22. To measure sequence lengths, we utilize the `tiktoken` tokenizer with the `cl100k_base` encoding scheme to count tokens for queries and corpus documents.

## H TRAINING DETAILS

For the query encoder, we use the final hidden state of the [CLS] token as the sentence embedding. For the document encoder, we append an [EOS] token to the input sequence and use its output hidden state as the document embedding. The maximum input length for both queries and documents is set to 512 tokens.

We summarize the training configurations in Table 23 and Table 24. For memory efficiency, we enable gradient checkpointing and use DeepSpeed Stage 0. For models up to 4B parameters, we train in `fp16`, while for the 8B model we switch to `bf16` to ensure stability. All document encoders are fine-tuned with LoRA (rank 32, $\alpha = 64$). For all symmetric architectures of baselines and ours, models are fine-tuned for two epochs.

In our asymmetric architecture, both query and document encoders are first initialized by one epoch of fine-tuning. For Stage I, we align query and document embeddings using $8.4$M pairs of query alignment data for one epoch. For Stage II, we further fine-tune for one epoch to ensure comparability with other baselines. We apply the same learning rate ($1 \times 10^{-4}$) to both query and document encoders, as we observed that asymmetric learning rates led to performance degradation.

## I ANNOTATION PROMPTS

Table 25 and Table 26 present the prompts templates for MedTEB construction. Table 27 shows prompt for our training data annotation.

## J LLM USAGE

Large Language Models (LLMs) were used in two aspects of this work: (1) LLMs were employed for preliminary annotation during the construction of training data and the MedTEB benchmark,

with all data anonymized and curated; (2) LLMs were used as a writing aid to polish the manuscript. All research ideas, model design, experiments, and analysis were conducted by the authors.

Table 15: An example of false negatives in MedicalRetrieval. This example shows the *topic intensity* phenomenon in medical domain: certain diseases or drugs generate a large volume of semantically similar queries and answers.

**Query**

感冒发烧一起来怎么办

*What should I do if I have a cold and fever at the same time?*

**Positive**

你好，应该是流行感冒吧，典型流感，急起高热，全身疼痛，显著乏力，呼吸道症状较轻。颜面潮红，眼结膜外眦充血，咽充血，软腭上有滤泡。具体吃药建议咨询医生。期间注意：多饮开水，多食新鲜的蔬菜、瓜果。饮食宜清淡，多食易消化，且富于营养及富含维生素的食物，如稀饭、豆浆、面条、水果等。窗子经常开下来通通风。一些简单食疗：1.生姜20克、大蒜头5～6瓣、红糖适量。用法：水煎服。主治：流行性感冒初起，头痛，怕冷发热，无汗，伴有恶心者。说明：流行性感冒是一种急性呼吸道传染病，表现为急起高热，全身疼痛，乏力，呼吸道症状如咽干喉痒，干咳等，胃肠道症状如恶心呕吐、腹泻水样便等。2.冬瓜粥 粳米50克。将冬瓜适量切成小块，与米同煮，粥熟即可食用。此粥对病毒型流行性感冒病人有效。3.葱白500克、大蒜250克。用法：上药切碎加水2000毫升煎煮。日服3次，每次250毫升，连服2～3天。愿早日康复！

*Hello, it sounds like you have the flu. Typical symptoms include sudden high fever, body aches, significant fatigue, and mild respiratory symptoms. You may also have facial flushing, conjunctival injection, pharyngeal congestion, and follicles on the soft palate. Please consult a doctor for specific medication advice. During this time, drink plenty of water and eat more fresh vegetables and fruits. Keep your diet light and easy to digest, focusing on nutritious and vitamin-rich foods like porridge, soy milk, noodles, and fruits. Ventilate your room regularly by opening windows. Here are some simple home remedies: 1. 20 grams of fresh ginger, 5–6 cloves of garlic, and an appropriate amount of brown sugar. Decoct in water and take orally. This is for the early stages of influenza with headache, chills, fever, no sweating, and nausea. Influenza is an acute respiratory infectious disease characterized by sudden high fever, body aches, fatigue, and respiratory symptoms like sore throat and dry cough. It may also cause gastrointestinal symptoms like nausea, vomiting, and watery diarrhea. 2. Winter melon porridge: 50 grams of japonica rice. Cut an appropriate amount of winter melon into small pieces and cook with rice. This porridge is effective for patients with viral influenza. 3. 500 grams of green onion whites and 250 grams of garlic. Chop the ingredients and decoct in 2000 milliliters of water. Take three times a day, 250 milliliters each time, for 2–3 days. Hope you recover soon!*

**False Negative 1 (Labeled as negative, but annotated as positive by LLM)**

建议口服抗病毒药物和感冒冲剂试试.有炎症还是应该加上抗生素.口服药物不见效的，建议输液治疗为好。在当地医生指导下使用。发烧用退热贴

*It is suggested to try oral antiviral medications and cold granules. If there is an infection, antibiotics should be added. If oral medications are not effective, it is recommended to consider intravenous therapy. This should be done under the guidance of a local doctor. For fever, you can use fever patches.*

**False Negative 2**

感冒发烧是临床上最常见的疾病和症状，具体吃药要根据具体的症表现以及病人身体状况而定。如果是儿童出现感冒发烧的情况一般选择以单药为主，出现发烧时主要可选择对乙酰氨基酚或者布洛芬口服液来进行治疗；如果还有其他的症状，比如出现鼻塞流涕，可以使用氨咖黄敏颗粒。如果是成人感冒发烧，一般多选择复合剂型，比如酚麻美敏片或者复方氨酚烷胺等。如果持续发烧不退，要及时完善血液分析和胸片检查排除并发肺炎的可能。

*A cold with fever is one of the most common illnesses and symptoms clinically. The specific medication should be determined based on the specific symptoms and the patient's physical condition. For children with a cold and fever, monotherapy is usually chosen. For fever, acetaminophen or ibuprofen oral suspension can be used for treatment. If there are other symptoms, such as nasal congestion and runny nose, pheniramine and caffeine granules can be used. For adults with a cold and fever, compound formulations are generally preferred, such as phenylephrine, dextromethorphan, and acetaminophen tablets, or compound paracetamol and amantadine. If the fever persists, it is important to promptly complete blood tests and chest X-rays to rule out the possibility of pneumonia.*

**False Negative 3**

你好，建议口服抗病毒药物和感冒冲剂试试.即使是病毒性感冒也容易继发细菌感染,所以最好还是应该加上抗生素口服.建议口服药物不见效的,建议输液抗炎治疗为好.因为还是输液血药浓度更高见效更快更好啊.有痰的加上鲜sd竹沥口服试试.发烧还需要适当加上额外的退烧药物.一般需要7-10天才能治愈的.最好还是看医生啊

*Hello, it is suggested to try oral antiviral medications and cold granules. Even viral colds can easily lead to secondary bacterial infections, so it is better to add oral antibiotics. If oral medications are not effective, it is recommended to consider intravenous anti-inflammatory treatment, as it provides higher blood drug concentration and faster results. For those with phlegm, you can try adding fresh bamboo extract orally. Fever also requires the addition of extra antipyretic drugs. It usually takes 7-10 days to recover. It is best to see a doctor.*

Table 16: MedTEB-Retrieval example.

**Query**
肾结石如何判断是酸性还是碱性结石？
*How to ascertain whether a renal calculus is acidic or alkaline in composition?*

**Positive example**
咋知道肾结石是有酸性碱性引起
病情分析：一般通过尿检判断肾结石是酸性的还是碱性的，可以到本地正规医院做尿液，酸碱度检查也可以观察一下pH值的变化，然后再明确一下尿液的酸碱度。如果怀疑身体有肾结石的症状，可以到正规医院做影像学检查检查一下大小。如果结石比较大的话，一定要及时到医院做激光碎石治疗。
*How can I tell whether a kidney stone is related to acidic or alkaline urine? Clinical assessment: In general, urinalysis is used to determine whether a renal calculus is associated with acidic or alkaline urine. You can have a urine pH test at a qualified local hospital and monitor the pH value to establish urinary acidity or alkalinity. If kidney stone symptoms are suspected, undergo imaging studies to assess the stone size. If the calculus is relatively large, timely laser lithotripsy is recommended.*

Table 17: MedTEB-Rerank example.

**Query**
耳鸣需要吃什么药？
*Which medications are indicated for tinnitus?*

**Positive example**
耳鸣的药有哪些
病情分析：耳鸣常用的药物有，1.盐酸氟桂利嗪胶囊、尼莫地平等，用于改善耳蜗的供血，扩张耳蜗血管。2. 三磷酸干、辅酶A、甲钴胺等，用于改善耳道的代谢功能，可以促进耳部的新陈代谢，清理耳道杂质。3.卡马西平、路硝西泮等，用于抗惊厥，能够缓解耳朵受到刺激造成的耳鸣。4. 抗生素、红霉素、万古霉素等，这些药物含有非类固醇消炎药物，可以给耳道涂抹起到消炎的作用，以此来缓解耳鸣。
*What medications are available for tinnitus? Clinical assessment: Commonly used drugs include flunarizine hydrochloride capsules and nimodipine to improve cochlear perfusion by dilating cochlear vessels; adenosine triphosphate (ATP), coenzyme A, and mecobalamin (methylcobalamin) to enhance metabolic function of the auditory pathway, promote aural metabolism, and help clear debris from the ear canal; carbamazepine and clonazepam as anticonvulsants to relieve tinnitus triggered by neural irritation; and antibiotics such as erythromycin and vancomycin, as well as nonsteroidal anti-inflammatory agents, which can be applied to the ear canal for anti-inflammatory effects to help alleviate tinnitus.*

**Negative example**
吃补肾的药怎么耳鸣呢
病情分析：患者是由于肾阴亏虚而引起的上火症状，进而导致患者出现耳鸣。首先，患者应该服用一些滋阴补肾的药物来进行补肾，比如六味地黄丸或者知柏地黄丸。等到患者的肾虚得到一定的恢复之后，耳鸣的症状也会逐渐的消失。另外，患者可以搭配服用一些清热泻火的药物来进行治疗。
*Why would taking kidney-tonifying medicine lead to tinnitus? Clinical assessment: From a traditional Chinese medicine perspective, the patient's tinnitus is due to kidney-yin deficiency with endogenous heat, which precipitates tinnitus. It is advisable to use yin-nourishing, kidney-tonifying formulas such as Liuwei Dihuang Wan or Zhibai Dihuang Wan. As the kidney deficiency improves, the tinnitus should gradually resolve. In addition, heat-clearing and fire-purging agents can be used concomitantly.*

Table 18: MedTEB-STS example.

**Sentence1**
碳酸氢钠片是否会引起头皮痒
*Do sodium bicarbonate tablets cause scalp itching?*

**Sentence2**
服用小苏打片是否可能导致头皮发痒？
*Could taking baking soda tablets lead to an itchy scalp?*

Table 19: Statistics of the MedTEB Retrieval test set.

| Dataset | Split | # Queries | Avg. Q. Len. | # Corpus | Avg. Doc. Len. | Avg. Pos. |
|---|---|---|---|---|---|---|
| Retrieval | Test | 734 | 20.68 | 229,457 | 470.90 | 8.43 |

Table 20: Statistics of MedTEB Rerank and CMedQA-v1/v2 Rerank test sets.

| Dataset | Split | # Queries | Avg. Q. Len. | Avg. Docs/Q | Avg. Doc. Len. | Avg. Pos. |
|---|---|---|---|---|---|---|
| Rerank | Test | 1,128 | 18.52 | 27.83 | 502.75 | 7.83 |
| CMedQA-v1 | Test | 1,000 | 75.58 | 100.00 | 143.03 | 1.93 |
| CMedQA-v2 | Test | 1,000 | 66.98 | 100.00 | 135.99 | 1.91 |

Table 21: Statistics of the MedTEB STS test set.

| Dataset | Split | # Queries | Avg. Q. Len. | Pos. Labels | Pos. Ratio |
|---|---|---|---|---|---|
| STS | Test | 5,000 | 35.45 | 2,396 | 47.92% |

Table 22: Statistics of MedTEB train sets.

| Dataset | Split | # Queries | Avg. Q. Len. | # Corpus | Avg. Doc. Len. |
|---|---|---|---|---|---|
| Retrieval/Rerank | Train | 20,000 | 21.24 | 229,457 | 470.90 |
| CMedQA | Train | 50,000 | 66.76 | 196,902 | 134.22 |
| STS | Train | 10,000 | 23.52 | 24,906 | 29.95 |

Table 23: General training hyperparameters. RetroMAE and Unsupervised are applied for Medical-Embedder-base pretraining. Fine-tuning applies to all other stages unless otherwise specified.

| Configuration | RetroMAE | Unsupervised | Fine-tuning |
|---|---|---|---|
| Optimizer | AdamW | AdamW | AdamW |
| Peak learning rate | $2 \times 10^{-4}$ | $1 \times 10^{-4}$ | $1 \times 10^{-4}$ |
| Warmup ratio | 0.0 | 0.05 | 0.05 |
| LR scheduler | linear decay | linear decay | linear decay |
| Global batch size | 384 | 19,200 | 640 |
| Epochs | 3 | 3 | 2 |

Table 24: Asymmetric training hyperparameters.

| Configuration | Stage I | Stage II |
|---|---|---|
| Optimizer | AdamW | AdamW |
| Peak learning rate | $1 \times 10^{-4}$ | $1 \times 10^{-4}$ |
| Warmup ratio | 0.05 | 0.05 |
| LR scheduler | linear decay | linear decay |
| Global batch size | 2,560 | 640 |
| Epochs | 1 | 1 |

Table 25: Prompt template for MedTEB Retrieval and Rerank tasks

Prompt:
This is a medical information retrieval task: given a medical query (Query), retrieve
documents (Passages)that can answer the question.

Given a medical query (Query) and {len(docs)} passages, your task is to rate the
relevance between the Query and each Passage.

Relevance scoring criteria:
S: The subject (e.g., disease name, drug name, inquiry target) and inten of Query and
Passage are fully consistent. The Passage can directly, completely, and correctly answer the Query.
A: The subject and intent of Query and Passage are consistent. The Passage contains
content that candirectly and correctly answer the Query.
B: The subject of Query and Passage is consistent, but the intent differs.
The Passage cannot directly answer the Query, but it is useful for inference.
C: The subject of Query and Passage is related, but the intent is inconsistent.
It can only partially match the Query from the text, but cannot answer the Query.
D: The subject and intent of Query and Passage are unrelated. Cannot answer the Query.

Notes:
1. Query and Passage are independent; there is no contextual relationship.
Do not infer or supplement the subject/intent of Query based on Passage.
2. If the Query is low-quality (e.g., missing subject, like "How to treat this disease?"),
the maximum relevance score for all Passages should not exceed B.
3. All Passages are independent; they are randomly ordered and have no contextual relationship.

Output format:
Your output must be a JSON object, containing only the required fields. The format is as follows:
{ "Passage-0": "A", "Passage-1": "C", ... }
Query and Passages are as follows:
- Query: {query}
{passages}
...
Remember: do not output any other content or explanation.
Your output must be only a JSON object with the required fields. Output:

Table 26: Prompt template for MedTEB STS tasks

Medical Query Rewriting Sample Generation (Positive and Negative Examples)

Task Objective: Your task is to generate one positive example and two negative examples based on a given original medical query and a set of synonyms.
You will receive a JSON object containing the following fields:
"origin": "Original medical term",
"replace": "Synonym medical term for replacement",
"query_pairs": { "origin": "Query sentence using the original term", "replace": "Query sentence using the replaced term" }

Generation Rules:
1. General Quality Standards (applicable to all outputs):
- Professional Expression: Use professional, fluent, and natural medical language.
- Medical Accuracy: Content must conform to medical knowledge and avoid ambiguity.
- Format Requirement: All outputs must be complete, fluent interrogative sentences.

2. Specific Sample Requirements:
- positive (Positive Example):
- Task: Optimize and rewrite the second query in query_pairs (the one containing the "replace" term).
- Intent: Must preserve the exact same intent as the original query.
- Terminology: Must use the term specified in the "replace" field.
- Constraint: Rewritten query length must be within ±30% of the original query's length.
- negative-1 (Negative Example 1):
- Task: Create a new query based on the topic of the original query, similar but distinctly different.
- Terminology: Must use the term specified in the "replace" field.
- Intent: Significantly alter the intent of the original query.
- negative-2 (Negative Example 2):
- Task: Create a new query based on the topic of the original query, similar but distinctly different.
- Terminology: Must use the term specified in the "origin" field.
- Intent: Significantly alter the intent of the original query (same rule as negative-1).

Output Format: Must be a JSON object containing only the following three fields. Do not add any extra explanations or comments.

Input: {input}
Output:

Table 27: Prompt template for training data annotation

This is a retrieval task in the Chinese medical domain, requiring classification of positive and negative documents based on the user's medical query and search engine returned documents.
You will receive data containing the following fields:
"query": User input in the medical domain.
"documents": A candidate document set containing multiple documents, some relevant and some irrelevant — capable or incapable of answering the query.
Your task is to identify "positive_document" and "negative_document" from the provided documents.
"positive_document": Relevant to the query; the document contains sentences that can answer the query.
"negative_document": Either relevant or irrelevant to the query, but the document content does NOT contain any sentence that can answer the query.
Please follow these guidelines:
- Both "positive_document" and "negative_document" must come from the candidate document set.
- "positive_document" and "negative_document" are mutually exclusive — no document overlap is allowed.
Output Requirements:
Example: {out_exam}.
Your output must always be ONLY a JSON object, containing ONLY document indices (e.g., "doc-1").
Do NOT include document content, explanations, or any additional text.
Input Data Format:
{"positive_document":["doc-1","doc-2"], "negative_document":["doc-3","doc-4"]}
Classify the documents in the input data according to the above rules, ensuring the output strictly follows the required format.
Output:

