# OpenReview forum: "MAR: Medical Asymmetric Retriever for Efficient Chinese Medical Dense Retrieval"
_ICLR.cc/2026/Conference — ICLR 2026 Conference Withdrawn Submission_

### Official Review · Reviewer_TpzX · 2025-10-20

**Soundness:** 3
**Presentation:** 2
**Contribution:** 2
**Rating:** 2
**Confidence:** 4

**Summary:**

This paper introduces MedTEB, a new benchmark for Chinese medical text embeddings covering three tasks—retrieval, reranking, and synonym STS—and proposes MAR (Medical Asymmetric Retriever), an asymmetric dual-encoder framework pairing a lightweight query encoder with a large document encoder.

The authors then design a two-stage optimization scheme (query alignment and joint fine-tuning) and demonstrate state-of-the-art performance on MedTEB while maintaining efficiency comparable to smaller BERT-style models.

**Strengths:**

1. This paper introduces a comprehensive benchmark (MedTEB) for Chinese medical embeddings with multiple tasks, addressing an underexplored but practically important domain.

2. This paper provides thorough empirical evaluation comparing MAR against a wide range of strong baselines (BGE, GTE, Qwen, etc.), demonstrating consistent efficiency–accuracy improvements.

**Weaknesses:**

1. The proposed MAR framework is primarily a combination of existing asymmetric retrieval and distillation methods (e.g., KALE, HotelMatch), with only minor architectural differences (removing projection layer, using MSE + InfoNCE). The methodological innovation is minimal.

2. Annotation reliability not verified: LLM-based labeling is claimed to be high-quality, but no inter-LLM agreement or human validation statistics are reported.

3. Improvement margins are modest (≈1.3–1.5%), and the methods rely heavily on proprietary LLMs for annotation. The authors also did not  compare with baselines using more parameters such as Qwen-3-embedding-4b, GTE etc.

4. It would be better to also evaluate over different retrieval-based applications such as chinese medical QA.

5. Missing references on medical retrievers:

> [1] Xu et al. "BMRetriever: Tuning Large Language Models as Better Biomedical Text Retrievers." Proceedings of the 2024 Conference on Empirical Methods in Natural Language Processing. 2024.
> [2] Jin et al. "Medcpt: Contrastive pre-trained transformers with large-scale pubmed search logs for zero-shot biomedical information retrieval." Bioinformatics 39.11 (2023): btad651.
> [3] Li et al. "R2MED: A Benchmark for Reasoning-Driven Medical Retrieval." arXiv preprint arXiv:2505.14558 (2025).

**Questions:**

1. Can you quantify inter-LLM agreement rates or human verification accuracy to support the claim of “high-quality annotation”?

2. How sensitive is MAR to the choice of base document encoder (e.g., Qwen3 vs. BGE)?

3. How would MAR perform in non-medical Chinese retrieval tasks? Does the alignment generalize beyond the medical corpus?

---

> ### Comment · Reviewer_TpzX · 2025-11-20
> **The references I mentioned are real papers and not LLM generated**
>
> [1] Xu et al. "BMRetriever: Tuning Large Language Models as Better Biomedical Text Retrievers." Proceedings of the 2024 Conference on Empirical Methods in Natural Language Processing. 2024.  Link: https://aclanthology.org/2024.emnlp-main.1241/
>
> [2] Jin et al. "Medcpt: Contrastive pre-trained transformers with large-scale pubmed search logs for zero-shot biomedical information retrieval." Bioinformatics 39.11 (2023): btad651.  Link: https://arxiv.org/abs/2307.00589
>
> [3] Li et al. "R2MED: A Benchmark for Reasoning-Driven Medical Retrieval." arXiv preprint arXiv:2505.14558 (2025). Link: https://arxiv.org/abs/2505.14558

---

> ### Author Response · Authors · 2025-11-20
> **Official Comment by Authors [1/4]**
>
> Dear Reviewer,
> Thanks a lot for your insightful review. We appreciate it to clarify the questions within your review comments.
>
> > **W1:** The main method for MAR framework is a combination of existing asymmetric retrieval and distillation methods (e.g., KALE, HotelMatch), with only minor architectural differences (removing projection layer, using MSE + InfoNCE). The methodological innovation is minimal.
>
> While both of KALE and HotelMatch are great works, MAR significantly differs from both works in **core design, training framework and target modality**.
> - For KALE: KALE prunes layers from a BERT-based teacher as student encoder, and distill teacher embeddings to student via KL divergence. But MAR employs a **heterogeneous architecture** (encoder-only query encoder + decoder-only LLM document encoder), which is more generalizable. Also, MAR introduces a **two-stage training pipeline**, whose training design and objective are different from KALE.
> - For HotelMatch: HotelMatch uses single-stage contrastive learning for multimodal (image–text) retrieval. MAR designs a **text-only**, **two-stage alignment strategy** specifically for latency-sensitive passage retrieval. The objectives, modalities, and training procedures are therefore fundamentally different.
>
> Table 1 (also found in Table 7 and Table 5 in our paper) shows that KALE and HotelMatch underperform on LLM-based text embeddings, while MAR achieves substantial gains, demonstrating limited overlap.
> To the best of our knowledge, **MAR is the first open-sourced asymmetric text embedding models with an LLM-based document encoder**, offering a practical recipe for deploying large embedders in real-world search.
>
> Table 1: Comparison with KALE and HotelMatch.
> | Model       | Params(Q/D)  | Retrieval | Rerank | Avg   |
> |-------------|--------------|-----------|--------|-------|
> | KALE        | 302.8M/4.02B | 42.67     | 67.42  | 55.05 |
> | HotelMatch  | 305M/4.02B   | 35.34     | 66.79  | 51.07 |
> | MAR-0.3B-4B | 305M/4.02B   | 55.91     | 72.84  | 64.38 |
>
> > **W2:** This paper relied on LLM-based annotation. However, this step may introduce additional bias and inaccuracies and no inter-LLM agreement or human validation statistics are reported.
>
> - We report a fine-grained **inter-LLM agreement breakdown** among the three LLMs. Table 2 shows that **89.13%** of instances receive total agreement (all three labels identical), while the remaining 10.87% fall into partial agreement, showing great consistency.
> - The **Fleiss’ Kappa** score of LLM annotation is **0.731**, indicating substantial agreement among the models.
> - To verify clinical validity, 5,000 query–passage pairs from the MedTEB-Retrieval test set were independently re-annotated by **clinical experts**. Their labels agreed with the final LLM-voted labels in **93.3%** of cases, demonstrating the high accuracy and clinical relevance of our pipeline.
>
> Table 2: LLM annotation agreements statistics.
> | Llm aggreements statistics| Describe | ratio %  | count  |
> |-------------------------|------------|----------|--------|
> | Total aggrement          | 0 positive | 78.76    | 723907 |
> |                          | 3 positive | 10.37    | 95344  |
> | Partially aggrement     | 1 positive | 4.75     | 43630  |
> |                         | 2 positive | 6.12     | 56224  |
>
> > **W3-1:** Improvement margins are modest (≈1.3–1.5%)
>
> - Rather than building another giant LLM-based SOTA embedding model consuming unbearable latency, MAR targets the **latency-accuracy trade-off**. Figure 1 in our paper shows that, compared to LLM-based embedding models, MAR-0.3B-8B delivers comparable retrieval quality (2.5 % higher on nDCG@10), while running **~9.9× faster than gte-Qwen2-1.5B-Instruct**. Also, compared with Bert-scale embedding models, MAR achieves same QPS while boosts accuracy by **11.3 % over bge-large-zh and 6 % over gte-multilingual-base.**
>
> >  **W3-2:** and the methods rely heavily on proprietary LLMs for annotation.
>
> - MAR’s training pipeline itself **does not depend on proprietary LLMs**. LLM annotation is used solely to construct the MedTEB train/test labels, which is a common method in recent works [1,2,3].
> - Moreover, MAR’s **query-alignment stage requires only unsupervised (query–query and doc-doc) pairs**, significantly mitigating the usage of labeling for alignment, making MAR's training framework more practical.
>
>
>
> [1] Hard Negatives, Hard Lessons: Revisiting Training Data Quality  for Robust Information Retrieval with LLMs
>
> [2] MIRACL: A Multilingual Retrieval Dataset Covering 18 Diverse Languages
>
> [3] AIR-BENCH: Automated Heterogeneous Information Retrieval Benchmark

---

> ### Author Response · Authors · 2025-11-20
> **Official Comment by Authors [2/4]**
>
> > **W3-3:** The authors also did not compare with baselines using more parameters such as Qwen-3-embedding-4b, GTE etc.
>
> - We focused on the balance of latency-accuracy instead of surpassing all LLM-based models with Bert-based query encoder. Nevertheless, we do include strong baselines like Qwen3-embedding-0.6B and gte-Qwen2-1.5B-Instruct as our baselines in Table 3, Section 5.2, and MAR matches or exceeds their retrieval quality while running 9.9× faster.
>
> > **W4:** It would be better to also evaluate over different retrieval-based applications such as chinese medical QA.
>
> Thanks for the suggestion. Evaluating end-to-end RAG is indeed uncommon in text-embedding papers, and no standard Chinese-medical RAG benchmark yet exists. To nevertheless demonstrate practical utility, we built a controlled RAG experiment:
> - corpus: 362k train set of Huatuo-26M [4] from huggingface FreedomIntelligence/huatuo_encyclopedia_qa;
> - test set: 1k test set of FreedomIntelligence/huatuo_encyclopedia_qa.
> - generator ChatGLM-6B [5] (released May 2023, before Huatuo-26M, avoiding data leakage);
> - Baselines: Fine-tuned bge-large-zh-v1.5 (same training data as MAR).
> Table 3 shows that our MAR-0.3B-4B improves BLEU-4 and all ROUGE scores over the baseline, **leading to better end-to-end RAG performance**.
>
>  Table 3: Comparision of RAG results on Huatuo-26M.
> | Retriever             | Retrieved-Topk | Bleu-4   | Rouge-1   | Rouge-2  | Rouge-l   |
> |-----------------------|----------------|----------|-----------|----------|-----------|
> | No Retriever          | 0              | 6.44     | 13.76     | 7.34     | 13.36     |
> | MAR-0.3B-4B           | 1              | **7.52** | 14.69     | 8.29     | 14.09     |
> |                       | 2              | 7.32     | **15.57** | **8.39** | **14.75** |
> |                       | 3              | 7.21     | 14.21     | 8.03     | 13.89     |
> | BGE-large (finetuned) | 1              | 6.18     | 13.05     | 7.02     | 12.96     |
> |                       | 2              | 7.12     | 14.26     | 7.61     | 14.05     |
> |                       | 3              | 7.03     | 14.13     | 8.17     | 13.95     |
>
> > **W5:** Missing references on medical retrievers
>
> Thank you for pointing this out. We have now added the domain-specific retrieval works mentioned in the question in the revision of our paper.
>
> > **Q1:** Can you provide more case studies on the quality of LLM annotations?
>
> Please see our detailed quantification of inter-LLM agreements in the response to **W-2**. Besides, we manually analysed 100 randomly sampled partial-agreement cases and classified the causes. We categorize the reason of disagreement into three types: **Boundary relevance, LLM hallunicate and Noise query/documents**. Results in Table 4 reveal the challenge of medical domain QA relevance labeling, and also prove the necessity of our multi-LLM agreement labeling stragety.
>
> Table 4: Analysis of LLM disagreements reasons.
> | Reason                | Explains                                                                                             | Ratio % |
> |-----------------------|------------------------------------------------------------------------------------------------------|---------|
> | Boundary relevance    | Some queries or documents may carry mutiple intents, making relevance labeling inherently difficult. | 50      |
> | Llm hallunicate       | Limited medical knowledge causes models to misjudge complex terms or relationships.                 | 38      |
> | Noise query/documents | Low-quality or incomplete queries/documents mislead the LLM into incorrect relevance labels.         | 12      |
>
>
>
>
> [4] Huatuo-26M, a Large-scale Chinese Medical QA Dataset
>
> [5] ChatGLM: AFamily of Large Language Models  from GLM-130B to GLM-4 All Tools

---

> ### Author Response · Authors · 2025-11-20
> **Official Comment by Authors [3/4]**
>
> > **Q2**: How sensitive is MAR to the choice of base document encoder (e.g., Qwen3 vs. BGE)?
>
> MAR is **architecturally agnostic**: any BERT-scale query encoder can be paired with any LLM-based document encoder. We verify this along two dimensions of **parameter scale and model families**.
> - Parameter scale: Table 5 shows that MAR generalize well with different size (from Medical-Embedder-4B to Medical-Embedder-8B). With document encoder scaling, the performance of MAR improves (from 64.38 to 65.21).
> - We replace Qwen3 with Llama3.2-3B as document encoders. As shown in Table 5, MAR still aligns well, with asymmetric architecture drops <1 point versus symmetric Llama3.2-3B baseline (57.16 vs 58.08).  The lower absolute scores reflect Llama-3.2’s weaker Chinese-medical knowledge, not a failure of MAR's alignment framework. Thus the capability of document encoder is the primary performance ceiling of MAR.
>
> Table 5: Generalizability of MAR to document encoders.
> | Query Encoder         | Doc Encoder         | Params (Q/D) | Retrieval | Rerank | Average |
> |-----------------------|---------------------|--------------|-----------|--------|---------|
> | Medical-Embedder-base | -                   | 305M/305M    | 54.16     | 69.63  | 61.90   |
> | Medical-Embedder-base | Medical-Embedder-4B | 305M/4.02B   | 55.91     | 72.84  | 64.38   |
> | Medical-Embedder-base | Medical-Embedder-8B | 305M/8.19B   | 56.75     | 73.67  | 65.21   |
> | Medical-Embedder-4B   | -                   | 4.02B        | 56.85     | 73.26  | 65.06   |
> | Medical-Embedder-8B   | -                   | 8.19B        | 57.79     | 73.47  | 65.63   |
> | Llama-3.2-3B-ft       | -                   | 3B           | 48.8      | 67.35  | 58.08   |
> | Medical-Embedder-base | Llama-3.2-3B-ft     | 305M/3B      | 47.64     | 66.68  | 57.16   |
>
> > **Q3:** How would MAR perform in non-medical Chinese retrieval tasks? Does the alignment generalize beyond the medical corpus?
>
> We conduct experiments on MMarco [6], which is a translated Chinese version of MSMarco [7] widely used to examine the ability of text embedding models in the general domain. Results in Table 6 explicitly show the **generalization ability beyond the medical corpus**: remove either stage of the asymmetric alignment training stage leads to a significant performance drop. With document encoder scaling, the performance has a consistent gain.
>
> Table 6: Results on general domain.
> | Mmarco                    | nDCG@10 | nDCG@20 | Recall@10 | Recall@20 |
> |---------------------------|---------|---------|-----------|-----------|
> | w/o query align (4B)      | 11.25   | 15.12   | 18.46     | 20.09     |
> | w/o joint finetuning (4B) | 15.58   | 16.35   | 21.2      | 24.19     |
> | Full Model (4B)           | 41.34   | 42.99   | 53.7      | 60.17     |
> | Full Model (8B)           | 46.67   | 48.42   | 59.48     | 66.25     |
>
> Thanks again for your careful review. Please let us know if you have further questions, we would be glad to provide comprehensive and detailed responses.
>
>
>
> [6] C-Pack: Packed Resources For General Chinese Embeddings
>
> [7] MSMARCO:AHumanGeneratedMAchine  Reading COmprehension Dataset

---

> > ### Author Response · Authors · 2025-11-26
> > **Official Comment by Authors [4/4]**
> >
> > Dear reviewer,
> >
> > We sincerely thank you for your insightful comments! Given that 7 days remain in the discussion period, we wish to ensure that all concerns have been fully addressed. We look forward to any additional feedback and are ready to answer further questions!

---

### Official Review · Reviewer_Qri6 · 2025-10-29

**Soundness:** 2
**Presentation:** 2
**Contribution:** 2
**Rating:** 4
**Confidence:** 3

**Summary:**

This paper introduces MedTEB, a high-quality benchmark for Chinese medical text retrieval, reranking, and synonym understanding, addressing limitations in existing datasets. To enable both strong accuracy and low latency, the authors propose MAR, an asymmetric retriever that combines a lightweight query encoder with a powerful offline document encoder, aligned through a two-stage training strategy. MAR achieves state-of-the-art performance on MedTEB while remaining efficient for real-time medical retrieval and RAG applications.

**Strengths:**

+ By leveraging the capabilities of LLMs, this paper validates the issues and shortcomings present in current benchmarks, thereby uncovering noise-related problems in the data.
+ The authors introduce an asymmetric retrieval framework and demonstrate its effectiveness on this benchmark, achieving state-of-the-art performance.
+ The research objectives are clearly defined and effectively addressed.

**Weaknesses:**

Perhaps due to space constraints, several issues remain unclear to me:
+ For instance, in which specific aspects do the existing benchmarks fall short, and to what extent are these deficiencies present? Can experiments based solely on LLMs sufficiently support the claim made in the original text: "However, the retrieval tasks suffer from annotation noise and false negatives, leaving only the reranking tasks relatively reliable"?
+ The description of the benchmark construction process in this paper lacks sufficient detail. It is unclear whether many of the construction details, such as data distribution, adhere to benchmark construction standards.
+ The proposed model employs an asymmetric structure, yet why are all the baselines symmetric? The selection of baselines does not appear to be comprehensive.

**Questions:**

See above

---

> ### Author Response · Authors · 2025-11-20
> **Official Comment by Authors [1/4]**
>
> Dear reviewer,
> Thanks for your insightful review of our paper, and we appreciate it to clarify the questions within your review comment.
>
> > **W1-1:** For instance, in which specific aspects do the existing benchmarks fall short, and to what extent are these deficiencies present?
>
> Existing Chinese medical text retrieval benchmarks (MedicalRetrieval and CmedqaRetrieval) suffers from severe **False Negative problem**: many real-positive passages are marked as negative，resulting in a large number of mislabelled documents.
> - Experiments based on GPT-4o re-labeling on Table 13, Appendix E in our paper show the severity of the problem. For CmedqaRetrieval, **only 1.42 documents are labeled as positive per query on average, while GPT-4o labels ~20 documents as positive** per query, indicating a severe mislabelling issue. Table 15, Appendix E in our paper further presents examples of false negatives.
> - The correctness of relevance labels is **pivotal for both training and evaluating text-embedding models**. Recent studies have consistently highlighted the impact of false negatives: [1] found that false negatives in training corpus severely degrades the performance of embedding models, [2] proposes a multi-step pipeline combining native-speaker annotation with automatic validation to secure label accuracy. [3] introduces a complex quality control framework to correct the false relevance labels.
>
> Table13:LLM re-annotation on open-source medical retrieval benchmarks.
> | Benchmark        | Origin Pos | LLM-Labeled as Pos | False Positive | False Negative |
> |------------------|------------|--------------------|----------------|----------------|
> | MedicalRetrieval | 0.81       | 9.11               | 0.26           | 8.56           |
> | CmedqaRetrieval  | 1.42       | 19.94              | 0.46           | 18.98          |
>
> > **W1-2:** Can experiments based solely on LLMs sufficiently support the claim made in the original text: "However, the retrieval tasks suffer from annotation noise and false negatives, leaving only the reranking tasks relatively reliable"?
>
> We validated LLM-generated labels by human assessors re-judging 500 randomly sampled q-p pairs that the LLM labeled as false negatives. **The consistency rate between model and human annotation reached 92%**, sufficiently supporting our claim that current Chinese medical retrieval benchmarks severely suffer from false negatives noise.
>
>
> [1] Hard Negatives, Hard Lessons: Revisiting Training Data Quality  for Robust Information Retrieval with LLMs
>
> [2] MIRACL: A Multilingual Retrieval Dataset Covering 18 Diverse Languages
>
> [3] AIR-BENCH: Automated Heterogeneous Information Retrieval Benchmark

---

> ### Author Response · Authors · 2025-11-20
> **Official Comment by Authors [2/4]**
>
> > **W2:** The description of the benchmark construction process in this paper lacks sufficient detail. It is unclear whether many of the construction details, such as data distribution, adhere to benchmark construction standards.
>
> We sincerely apologize that, due to paper length limitations, details of benchmark construction process were insufficient in the main body. We describe the **detailed construction process, dataset statistics, query distribution and LLM-labeling validation** as below.
> 1. As present in Section 3.1 and Appendix B in our paper, the **construction details** are as follows:
> - **Retrieval**:
>     Given a query $q_i \in \mathcal{Q}$, we use gte-multilingual-base, bge-m3, Conan-embedding-v1 to retrieve and gather a candidate pool of top-500 documents $\mathcal{D}\_i $.
>     Three strong LLMs, DeepSeek-V3, Doubao-1.5-Pro and GPT-4o then rated each $(q_i, d_j)$ pair on a 5-point relevance scale. To ensure label quality, a document was retained as positive only when all three LLMs agreed, while pairs with partial agreement (only 1 or 2 agreements) were discarded.
>     The final retrieval dataset comprise a query set $\mathcal{Q}$, a refined corpus $\mathcal{D}' \subseteq \mathcal{D}$, and relevance labels $\mathcal{R} = \{(q_i, d_j, y_{ij}) \mid y_{ij} \in \{0,1\}\}$
> - **Rerank**:
>     We use the same multi-LLM consensus annotation as in Retrieval. For each query $q_i \in \mathcal{Q}$, we derive positives $P_i=\{d_j\in\mathcal{D}': y_{ij}=1\}$ and negatives $N_i=\{d_j\in\mathcal{D}': y_{ij}=0\}$.
>     The final reranking dataset is a collection of triplets $\mathcal{T}_{\textbf{Rerank}}=\{(q_i,\mathcal{P}_i,\mathcal{N}_i)\}$, where $\mathcal{P}_i$ is a list sampled from $P_i$ and $\mathcal{N}_i$ is a list sampled from $N_i$.
> - **STS**:
>     We first build a medical synonym dictionary with domain experts. For each $q_i \in \mathcal{Q}$, GPT-4o generates three sentences: a positive $s^+\_i$ (synonym substitution with semantics preserved), a hard negative $s^-\_{i,1}$ (synonym substitution with semantics changed), and an easy negative $s^-\_{i,2}$ (no synonym substitution with semantics changed).
>     We then sample $ s_i \in \{s_i^{+}, s_{i,1}^{-}, s_{i,2}^{-}\} $ and pair it with $ q_i $ to form $ (q_i, s_i, y_i) $, where $ y_i=\mathbf{1}[\,s_i=s_i^{+}\,]\in\{0,1\} $.
>     The dataset is $ \mathcal{T}_{\text{STS}}=\{(q_i,s_i,y_i)\} $, evaluating fine-grained synonym understanding.
> 2. **Statistics** of MedTEB datasets: Please refer to Appendix G in the revision version of our paper for the statistics of MedTEB datasets.
>
>
> 3. **Query distribution**: We further employed GPT-4o to classify all queries into seven intent-based categories. The resulting distribution, shown in Table 1, demonstrates the broad diversity of query types in our dataset.
>
> Table 1: Query distribution of MedTEB test datasets.
> | Task   | Description           | Retrieval | Rerank | STS    |
> |--------|-----------------------|-----------|--------|--------|
> | CLIN   | Clinical Decisions    | 47.96%    | 49.73% | 39.38% |
> | PREV   | Prevention & Wellness | 19.21%    | 20.39% | 13.52% |
> | MED    | Medication Use        | 18.94%    | 22.61% | 44.32% |
> | OBG    | Women & Child Health  | 3.54%     | 3.20%  | 1.04%  |
> | NAV    | Navigation            | 2.23%     | 1.10%  | 0.52%  |
> | DERIV  | Medical Meta          | 7.08%     | 2.30%  | 0.56%  |
> | NONMED | Unclassifiable        | 1.04%     | 0.71%  | 0.66%  |
>
> 4. **LLM-labeling validation**: We validate the professionalism of LLM-annotations as below.
> - **Fleiss’ Kappa** on the relevance annotations produced by the three LLMs is **0.731**, indicating substantial agreement among the models.
> - A **clinical expert** independently re-annotated a random sample of 5,000 query–document pairs from the MedTEB-Retrieval test set. The agreement rate is **93.3%**.

---

> ### Author Response · Authors · 2025-11-20
> **Official Comment by Authors [3/4]**
>
> > **W3:** The proposed model employs an asymmetric structure, yet why are all the baselines symmetric? The selection of baselines does not appear to be comprehensive.
>
> We appreciate the reviewers’ concern. However, **no directly comparable asymmetric baseline exists for the main-results in Section 5.2**.
> At present, every top entry on the official MTEB leaderboard is a symmetric dual-encoder, and asymmetric architectures still lack sufficient exploration and development. To the best of our knowledge, **we are the first open-sourced asymmetric text embedding models with a LLM-based document encoder**.
> To still demonstrate the effectiveness of MAR's asymmetric training framework, we replicated and compared other asymmetric works as ablation studies in Section 5.3. Results in Table 7 in our paper demonstrate the effectiveness of our two-stage alignment framework.
>
> Table 7: Ablation study on other asymmetric baselines.
> | Model          | Asym | Params(Q/D)  | Retrieval | Rerank | Avg   |
> |----------------|------|--------------|-----------|--------|-------|
> | KALE           | ✓    | 302.8M/4.02B | 42.67     | 67.42  | 55.05 |
> | Wang&Lyu(2023) | ✓    | 302.8M/4.02B | 39.99     | 66.26  | 53.13 |
> | ScalingNote    | ✓    | 305M/4.02B   | 34.81     | 64.17  | 49.49 |
> | MAR-0.3B-4B    | ✓    | 305M/4.02B   | 55.91     | 72.84  | 64.38 |
>
> Thanks again for your careful review. If you have further questions, we would appreciate it to provide comprehensive and detailed responses.

---

> > ### Author Response · Authors · 2025-11-26
> > **Official Comment by Authors [4/4]**
> >
> > Dear reviewer,
> >
> > We truly appreciate your insightful feedback! There are 7 days remaining in the discussion phase, and we would like to confirm that all concerns have been adequately resolved. Please let us know if anything else needs clarification, and we are glad to answer further questions!

---

### Official Review · Reviewer_7fKN · 2025-10-31

**Soundness:** 3
**Presentation:** 3
**Contribution:** 2
**Rating:** 4
**Confidence:** 4

**Summary:**

The research seeks to address the challenge in medical information retrieval where existing search systems must tradeoff being fast but inaccurate or accurate but slow, making them impractical for real-time clinical use. This problem is stated as especially acute for Chinese medical text, where testing benchmarks are incomplete and specialized tools are lacking. Researchers propose two contributions: MedTEB, a comprehensive benchmark using real medical queries and multi-AI verification systems, and MAR (Medical Asymmetric Retriever), an innovative two-part search architecture that separates the search process into complementary components.

MARs uses a small "query encoder" to process searches in real-time paired with a large, powerful "document encoder" that pre-processes medical documents offline. This approach—trained through a three-stage process of independent training, alignment, and joint fine-tuning—eliminates the traditional speed-accuracy trade-off. The results claim to demonstrate that MAR achieves the high accuracy of larger models while maintaining the processing speed of smaller ones.

**Strengths:**

The work seeks to address the tradeoffs between speed, computational cost and accuracy often encountered in document retrieval systems. This is an important (albeit well-established) challenge in this area. The application to Chinese text documents and a curated, benchmark dataset of such documents along with the code is also a clear contribution of this work. Also, and in general, the paper is well-written and technically sound.

**Weaknesses:**

As noted above, the challenge itself is not novel, per se, nor is the strategy of preprocessing and creating an index offline and using the index at run time (hence my rating). Perhaps more detail on what is special about this specific manifestation of the problem and on the 2-tier solution could increase the novelty of this work. Also, once the dataset and the code has matured and the community has built on it, this work will have much greater impact. The authors should be encouraged to pursue this direction.

**Questions:**

Could the authors provide a better characterization of the novelty of both the problem and the solution?

---

> ### Author Response · Authors · 2025-11-20
> **Official Comment by Authors [1/2]**
>
> Dear reviewer,
> Thanks for your acknowledgement of our paper. We are highly appreciate the opportunity to clarify the questions within your review comments.
>
> > Could the authors provide a better characterization of the novelty of both the problem and the solution?
>
> Here is a comprehensive characterization of the novelty from both **new problems and new solutions** introduced by this paper.
> - Problems:
>     1. **New challenge brought by LLM-based embedders**. We agree that, the tradeoffs between speed, computational cost and accuracy has long been a challenge encountered by many document retrieval systems, and many works have done on it (HNSW [1] and DistilBERT [2]， etc.). However, with recent development of LLM-based text embedding models, **billion-scale, decoder-only** LLM embedders now achieve new SOTA on MTEB [3] and C-MTEB [4] (please see official MTEB Leaderboard). It introduces a new challenge: their heavy size makes it infeasible to deploy in real world retrieval systems, limiting the utilization of the power of LLM-based embedding models on industrial search systems.
>     2. **New challenge brought by asymmetric architecture**. Independently trained text embedding models occupy unrelated embedding spaces. As shown in Table 1, the independently trained embedders (Independent init setting) behaves randomly in asymmetric architecture (scores 0 for Retrieval and 33.34 for Rerank, totally random results). Thus, how to design training objectives to better align two different text embedding models raises a non-trivial challenge.
>
> - Solutions:
>     1. **Architecture**: In dense passage retrieval, common practice is to get both query embeddings and document embeddings in a symmetric single model architecture. All top models on MTEB leaderboard remain symmetric, and the exploration of asymmetric architecture on text embedding models is still scarce. On aware that documents are always processed offline, and online queries are always short and easy, we propose an asymmetric architecture, with a lightweight embedding model serves online and a strong LLM-based embedder process offline. To our best knowledge, **we are the first open-source asymmetric text embedding models with a LLM-based document encoder**.
>     2. **Training Framework**: To better align 2 independently initialized embedding models, we propose a 2 stage asymmetric training framework: query-encoder alignment and joint finetuning. Our 2-stage training framework bridges embedding space of two heterogeneous models. As shown in Table 1, both components contribute to the final performance, leading to SOTA on MedTEB. MAR provides a new solution for applying LLM-based embedding models in latency-sensitive document retrieval systems.
>
> Table 1: Ablation study on training stages.
> | Setting              | Retrieval | Rerank | Avg   |
> |----------------------|-----------|--------|-------|
> | Independent init     | 0         | 33.34  | 16.67 |
> | w/o query align      | 35.34     | 66.79  | 51.07 |
> | w/o joint-finetuning | 42.69     | 68.28  | 55.49 |
> | Full model (4B)      | 55.91     | 72.84  | 64.38 |
>
>
> We believe these contributions advance both the practical deployment of LLM embedders and the theoretical understanding of asymmetric retrieval training.
>
> > Also, once the dataset and the code has matured and the community has built on it, this work will have much greater impact. The authors should be encouraged to pursue this direction.
>
> Thanks for your encouraging words! We are committed to continuously refining our open-source project and contributing it to the community, thereby fostering future development and providing a reliable solution for deploying LLM-based embedding models in production environments.
>
> Thank you again for your careful review. If you have any further questions, please let us know and we will be glad to provide thorough and detailed answers.
>
> [1] Efficient and robust approximate nearest  neighbor search using Hierarchical Navigable  Small World graphs
>
> [2] DistilBERT, a distilled version of BERT: smaller,  faster, cheaper and lighter
>
> [3] MTEB:Massive Text Embedding Benchmark
>
> [4] C-Pack: Packed Resources For General Chinese Embeddings

---

> > ### Author Response · Authors · 2025-11-26
> > **Official Comment by Authors [2/2]**
> >
> > Dear reviewer,
> >
> > Thank you again for your insightful comments! As the discussion remains open for another 7 days, we would like to ensure that all concerns have been addressed. We look forward to any additional feedback and are always ready to answer any new questions!

---

### Official Review · Reviewer_EYLG · 2025-11-02

**Soundness:** 3
**Presentation:** 2
**Contribution:** 4
**Rating:** 6
**Confidence:** 4

**Summary:**

This paper addresses key gaps in Chinese medical dense retrieval: the lack of high-quality benchmarks and the accuracy-efficiency trade-off of embedding models. It proposes two core contributions: (1) MedTEB, a Chinese Medical Text Embedding Benchmark covering 3 new tasks (Retrieval, Reranking, Synonym STS) and 2 existing high-quality datasets, constructed via multi-LLM consensus annotation to reduce noise; (2) MAR, a Medical Asymmetric Retriever with a lightweight online query encoder and a powerful offline document encoder, optimized via a two-stage framework (query alignment + joint fine-tuning) to achieve SOTA performance on MedTEB while maintaining low latency. The authors also commit to open-sourcing the benchmark, models, and code.

**Strengths:**

1. Fills the gap of reliable Chinese medical retrieval benchmarks: MedTEB solves annotation noise and false negatives in existing datasets (e.g., C-MTEB), providing a rigorous evaluation standard.
2. Practical asymmetric architecture: MAR balances real-time deployment needs (low-latency query encoder) and retrieval quality (powerful document encoder), breaking the accuracy-latency trade-off.
3. Significant domain value: Focuses on underdeveloped Chinese medical retrieval, supporting clinical decision support and medical RAG; open-source resources facilitate follow-up research.

**Weaknesses:**

1. Insufficient validation of LLM annotation professionalism: No clinical expert sampling verification or inter-annotator agreement metrics (e.g., Cohen’s Kappa) for MedTEB labels, risking "pseudo-professional" errors.
2. Unclear LLM annotation disagreement details: Fails to report the proportion of partially agreed samples or analyze disagreement causes, potentially compromising benchmark comprehensiveness.
3. Ambiguous document encoder pretraining: No information on the proportion of medical data in Qwen3’s pretraining corpus; lacks baseline comparisons (e.g., Llama3-8B) to verify Qwen3’s medical advantages.
4. Incomplete ethical/data compliance: No details on IRB approval for online user queries, specific anonymization steps, or exact public data sources, raising compliance concerns.
5. Lack of comparison with medical RAG methods: No performance comparison with domain-specific methods (e.g., Hyper-RAG), limiting competitiveness assessment.

**Questions:**

see weaknesses

**Details Of Ethics Concerns:**

1. Privacy risks: No confirmation of residual personal identifiers in "anonymized user queries" or detailed anonymization methods.
2. Legal gaps: Lack of IRB approval for user query data (potential human subjects) and unclear legal sourcing of public corpora, violating data protection laws.
3. Inadequate responsible disclosure: No information on annotator management (e.g., expert compensation) or safeguards against clinical misuse of released data.

---

> ### Author Response · Authors · 2025-11-20
> **Official Comment by Authors [1/4]**
>
> Dear Reviewer,
>
> Thank you for your insightful comments and for recognizing the contributions of our work. We appreciate the opportunity to address your concerns.
>
> > **W1:** Insufficient validation of LLM annotation professionalism: No clinical expert sampling verification or inter-annotator agreement metrics (e.g., Cohen’s Kappa) for MedTEB labels, risking "pseudo-professional" errors.
>
> We thank you for raising this important point. To assess both the reliability and clinical validity of the LLM-generated labels in MedTEB, we performed two complementary evaluations:
> - We computed **Fleiss’ Kappa** on the relevance annotations produced by the three LLMs in our pipeline. The Fleiss’ Kappa score is **0.731**, indicating substantial agreement among the models.
> - To assess the alignment with clinical expertise, a **clinical expert** independently re-annotated a random sample of 5,000 query–document pairs from the MedTEB-Retrieval test set. The agreement rate between expert annotations and our final LLM-voted labels is **93.3%**.
>
> > **W2:** Unclear LLM annotation disagreement details: Fails to report the proportion of partially agreed samples or analyze disagreement causes, potentially compromising benchmark comprehensiveness.
>
> In addition to the Fleiss’ Kappa score, we provide a detailed breakdown on the annotation agreement levels among the three LLMs, including the proportions of total agreement (all three LLMs assign the same label) and partial agreement (1–2 models positive). Table 1 shows that total agreement covers **89.13%** of the pairs, showing great consistency.
>
> Table 1: LLM annotation agreements statistics.
> | Llm aggreements statistics| Describe | ratio %  | count  |
> |-------------------------|------------|----------|--------|
> | Total aggrement          | 0 positive | 78.76    | 723907 |
> |                          | 3 positive | 10.37    | 95344  |
> | Partially aggrement     | 1 positive | 4.75     | 43630  |
> |                         | 2 positive | 6.12     | 56224  |
>
> Furthermore, we manually analysed 100 randomly sampled partial-agreement cases and classified the causes. We categorize the reason of disagreement into three types: **Boundary relevance, LLM hallunicate and Noise query/documents**. Results in Table 2 reveal the challenge of medical domain QA relevance labeling, and also prove the necessity of our multi-LLM agreement labeling stragety.
>
> Table 2: Analysis of LLM disagreements reasons.
> | Reason                | Explains                                                                                             | Ratio % |
> |-----------------------|------------------------------------------------------------------------------------------------------|---------|
> | Boundary relevance    | Some queries or documents may carry mutiple intents, making relevance labeling inherently difficult. | 50      |
> | Llm hallunicate       | Limited medical knowledge causes models to misjudge complex terms or relationships.                 | 38      |
> | Noise query/documents | Low-quality or incomplete queries/documents mislead the LLM into incorrect relevance labels.         | 12      |
>
> We will update these details in the revision of our paper for a more comprehensive validation of the LLM annotation professionalism.
>
> > **W3:** Ambiguous document encoder pretraining: No information on the proportion of medical data in Qwen3’s pretraining corpus; lacks baseline comparisons (e.g., Llama3-8B) to verify Qwen3’s medical advantages.
>
> While the Qwen3 technical report [1] does not explicitly disclose the proportion of Chinese medical data in its pretraining corpus, it states that **Qwen3 is extensively pretrained on large-scale, high-quality Chinese corpora**, which equips it with strong Chinese language understanding capabilities across a wide range of domains, including specialized ones like medicine.
> To empirically validate the suitability of Qwen3 as our document encoder, we conducted comparative experiments by fine-tuning several strong open-source baselines on the same medical retrieval task, including Llama-3-8B and Llama-3.2-3B [2]. Results in Table 3 show that both Qwen3-4B and Qwen3-8B significantly outperform Llama-3-8B and Llama-3.2-3B, demostrating Qwen3's superior Chinese medical text comprehension, and justifying its choice as the document encoder in MAR.
>
> Table 3: Comparision of document encoder baselines.
> | Base model   | Retrieve | Rerank | Avg   |
> |--------------|----------|--------|-------|
> | Llama-3.2-3B | 48.8     | 67.35  | 58.08 |
> | Qwen3-4B     | 56.85    | 73.26  | 65.06 |
> | Llama-3-8B   | 53.18    | 72.04  | 62.61 |
> | Qwen3-8B     | **57.79**    | **73.47**  | **65.63** |
>
>
> [1] Qwen3 Technical Report
>
> [2] The Llama 3 Herd of Models

---

> ### Author Response · Authors · 2025-11-20
> **Official Comment by Authors [2/4]**
>
> > **W4:** Incomplete ethical/data compliance: No details on IRB approval for online user queries, specific anonymization steps, or exact public data sources, raising compliance concerns.
>
> - **IRB approval**: Our research has been formally reviewed and approved by **the National Technology Ethics (Review) Committee, an IRB-equivalent ethics committee**.
> - **Specific anonymization steps**: All user queries and web documents were processed as follows.
>     1. Automated PII (personally identifiable information) Detection: We deployed an offline, locally hosted large language model to detect and mask potential PII, including names, locations, phone numbers, and ID numbers.
>     2. Rule-based Validation: After initial masking, we applied a rule-based validation module to scan residual digits, and keywords.
>     3. Human Checks: 1% of anonymized data were checked by human, and no re-identifiable content found.
> - **Data sources**:
>     - For queries: All user queries used in this study were collected from participants who explicitly opted into the user experience improvement program, with full consent for non-commercial research use only. All queries underwent rigorous anonymization and de-identification before any processing, ensuring no personally identifiable information (PII) could be traced back to individuals; no data were used outside this scope.
>     - For documents: All medical documents were crawled from publicly accessible, non-paywalled websites, primarily XunYiWenYao, which encourages web crawling for search engines (please see robots.txt of website). No login or authentication was required for access. We have removed all potential PII, and all the copyrights belong to the website.
>
> We have incorporated the above details into the ethics statement in our revised paper.
>
>
> > **W5:** Lack of comparison with medical RAG methods: No performance comparison with domain-specific methods (e.g., Hyper-RAG), limiting competitiveness assessment.
>
> After carefully analysis, we think **it is not suitable to compare MAR with other RAG systems** (e.g., Hyper-RAG) for following reasonings:
> 1. **Component VS Systems**: MAR is not a stand-alone RAG system, but a plug-and-play retriever component that can be inserted into any existing RAG pipeline to improve the retrieval stage. Also, most prior text embedding papers (e.g., GTE, BGE, E5) report only retrieval scores.[1,2,3,4].
> 2. **Evaluation Protocol Mismatch**: Hyper-RAG reports open-ended QA metrics, while MAR evaluates on retrieval tasks (nDCG or Recall etc.). The objectives are therefore not aligned.
>
> To nevertheless demonstrate MAR’s practical utility in a RAG setting, we built a Chinese medical RAG benchmark on Huatuo-26M [5] and report the results. The settings are as follows:
> - corpus: train split of FreedomIntelligence/huatuo_encyclopedia_qa in huggingface;
> - test set: 1k test set of FreedomIntelligence/huatuo_encyclopedia_qa.
> - generator ChatGLM-6B (released May 2023, before Huatuo-26M, avoiding data leakage);
> - Baselines: Fine-tuned bge-large-zh-v1.5 (same training data as MAR).
> Table 4 shows that replacing the baseline retriever with MAR yields consistent gains across BLEU-4 and all ROUGE variants, confirming that better retrieval translates into better end-to-end answers.
>
>  Table 4: End-to-end RAG performance on the Huatuo-26M QA test set (generator = ChatGLM-6B)
> | Retriever             | Retrieved-Topk | Bleu-4   | Rouge-1   | Rouge-2  | Rouge-l   |
> |-----------------------|----------------|----------|-----------|----------|-----------|
> | No Retriever          | 0              | 6.44     | 13.76     | 7.34     | 13.36     |
> | MAR-0.3B-4B           | 1              | **7.52** | 14.69     | 8.29     | 14.09     |
> |                       | 2              | 7.32     | **15.57** | **8.39** | **14.75** |
> |                       | 3              | 7.21     | 14.21     | 8.03     | 13.89     |
> | BGE-large (finetuned) | 1              | 6.18     | 13.05     | 7.02     | 12.96     |
> |                       | 2              | 7.12     | 14.26     | 7.61     | 14.05     |
> |                       | 3              | 7.03     | 14.13     | 8.17     | 13.95     |
>
> [1] M3-Embedding: Multi-Linguality, Multi-Functionality, Multi-Granularity  Text Embeddings Through Self-Knowledge Distillation
>
> [2] Qwen3 Embedding: Advancing Text Embedding and Reranking Through Foundation Models
>
> [3] C-Pack: Packed Resources For General Chinese Embeddings
>
> [4] Text Embeddings by Weakly-Supervised  Contrastive Pre-training
>
> [5] Huatuo-26M, a Large-scale Chinese Medical QA Dataset

---

> ### Author Response · Authors · 2025-11-20
> **Official Comment by Authors [3/4]**
>
> > **Ethics Concern 1**: Privacy risks: No confirmation of residual personal identifiers in "anonymized user queries" or detailed anonymization methods.
>
> We thank the reviewer for highlighting this critical issue. As detailed in our response to Question 4, we confirm that:
> 1. **No personal identifiers remain** in the released queries and corpus.
> 2. The three-step anonymisation pipeline was applied to all queries, and a **random 1 % human re-check found no re-identifiable content.**
>
> > **Ethics Concern 2**: Legal gaps: Lack of IRB approval for user query data (potential human subjects) and unclear legal sourcing of public corpora, violating data protection laws.
>
> Thank you for raising this important compliance point.
> Please refer to our response to Question 4 for detailed descriptions. We are confident that all our data construction process is **fully compliant with the data protection laws**.
>
> >  **Ethics Concern 3**: Inadequate responsible disclosure: No information on annotator management (e.g., expert compensation) or safeguards against clinical misuse of released data.
>
> Thank you for this essential reminder. Below we provide the missing details.
> - Clinical experts who re-annotated the 5 k sample work from a tertiary hospital that has a joint research agreement with our institution. Compensation was set at the institution-approved hourly rate for off-duty clinical consultants and was paid through the hospital’s contracted project budget; no annotator was personally solicited or paid directly by the authors.
> - The MedTEB dataset is released under **CC BY-NC-SA 4.0 license**; commercial use is explicitly forbidden. The model card will state 'No diagnostic use, for research purposes only'.
>
> We have expanded the ethics section to include these statements.
>
> Thank you once again for your careful review. We would be grateful for the opportunity to provide comprehensive and detailed responses if you have any further questions.

---

> ### Author Response · Authors · 2025-11-26
> **Official Comment by Authors [4/4]**
>
> Dear Reviewer,
>
> We again thank you for your insightful comments! With one week remaining for discussion period, we would like to confirm that we have fully addressed your concerns.
>
> We look forward to any further feedback and are ready to provide additional clarification whenever needed!

---

### Author Response · Authors · 2025-12-03

Dear Reviewers and Chairs,

Thank you for your careful and constructive feedback. We have addressed the reviewers' comments and updated the manuscript; the main changes and results are summarized below.

1. **MedTEB's annotation quality and statistics**
   - For LLM's annotation quality, we measured inter-annotator agreement using Fleiss’ κ, which reached 0.731 and reflects substantial consistency among annotators. We also include a clinical expert's validation, which achieved 93.3% agreement with the final LLM-voted labels on a set of 5k query–passage pairs. We added a fine-grained analysis of the 10.87% partial-agreement cases (boundary relevance 50%, LLM hallucination 38%, noisy input 12%).
   - We added detailed anonymization pipeline of MedTEB in Appendix B, and we included detailed dataset statistics in Appendix G.

2. **Ethics and data compliance**
   - IRB Approval: the study was reviewed by the National Technology Ethics (Review) Committee.
   - Anonymization: pipeline includes offline LLM detection, rule-based validation, and a 1% human check; no re-identifiable content was found.
   - Data sourcing: user queries come from an opted-in, non-commercial program; documents were crawled from public, non-paywalled sites that encourages web crawling for search engines. We plan to release the dataset under CC BY-NC-SA 4.0 with a research-only clause.

3. **Additional experiments**
   - Document-encoder sensitivity: MAR's document encoder transfers to Llama-3.2-3B with under a 1-point drop, supporting our architecture-agnostic design.
   - Cross-domain generalization: on MMARCO, MAR achieves 41.34 nDCG@10 versus 15.58 without joint fine-tuning, demonstrating MAR's generalization ability beyond the medical corpus.
   - End-to-end RAG: MAR-0.3B-4B improves BLEU-4 and all ROUGE scores over the baseline on Huatuo-26M QA, leading to better end-to-end RAG performance.

Please see our detailed response and the updated manuscript for more information. We look forward to any further questions from the Chairs!

---

### Note · Authors · 2026-01-04

I have read and agree with the venue's withdrawal policy on behalf of myself and my co-authors.